# Taming Diffusion for Dataset Distillation with High Representativeness

**Lin Zhao**[1]  **Yushu Wu**[1]  **Xinru Jiang**[1]  **Jianyang Gu**[2]  **Yanzhi Wang**[1]  **Xiaolin Xu**[1]  **Pu Zhao**[1]  **Xue Lin**[1]

## Abstract

Recent deep learning models demand larger datasets, driving the need for dataset distillation to create compact, cost-efficient datasets while maintaining performance. Due to the powerful image generation capability of diffusion, it has been introduced to this field for generating distilled images. In this paper, we systematically investigate issues present in current diffusion-based dataset distillation methods, including inaccurate distribution matching, distribution deviation with random noise, and separate sampling. Building on this, we propose D³HR, a novel diffusion-based framework to generate distilled datasets with high representativeness. Specifically, we adopt DDIM inversion to map the latents of the full dataset from a low-normality latent domain to a high-normality Gaussian domain, preserving information and ensuring structural consistency to generate representative latents for the distilled dataset. Furthermore, we propose an efficient sampling scheme to better align the representative latents with the high-normality Gaussian distribution. Our comprehensive experiments demonstrate that D³HR can achieve higher accuracy across different model architectures compared with state-of-the-art baselines in dataset distillation. Source code: https://github.com/lin-zhao-resoLve/D3HR.

## 1. Introduction

Driven by the scaling law, recent deep learning models have expanded in scale, demanding exponentially larger datasets for optimal training performance, which makes dataset maintenance costly and labor-intensive (Deng et al., 2009; Alexey, 2020). To achieve both computational and storage efficiency, dataset distillation generates a small distilled dataset to replace the full dataset for training (Wang et al., 2018; Loo et al.; Lee & Chung, 2024), while targeting comparable performance as the full dataset. In contrast to dataset pruning methods (selecting a subset from the full dataset) (Zhang et al., 2024; Sorscher et al., 2022; Tan et al., 2024), dataset distillation treats the distilled dataset as continuous parameters that are optimized to integrate information from the full dataset. This approach achieves better performance, especially with relatively high compression rates. (Du et al., 2024b; Sun et al., 2024; Yin et al., 2023).

Recent data distillation methods primarily focus on improving the efficiency of the distillation algorithm (Yin et al., 2023; Sun et al., 2024; Su et al., 2024; Gu et al., 2024). Among them, the diffusion-based methods achieve strong performance due to their remarkable generative capabilities. They capture the most informative features of the full dataset by extracting representative latents from the pre-trained Variational Autoencoder (VAE) latent space (Su et al., 2024; Gu et al., 2024). Thus, the architectural dependency during distillation is eliminated, enabling a one-time generation cost for training various model architectures. This strategy enhances the efficiency of distillation and significantly improves cross-architecture generalization.

However, the above diffusion-based methods may struggle to guarantee the representativeness of the distilled dataset. (i) **Inaccurate distribution matching.** This challenge arises from leveraging diffusion models to generate distilled datasets, relying on accurately modeling the VAE feature distributions for proper decoding. However, as shown in Figure 1a, the low-degree of normality[1] for the distribution in VAE latent space results in difficulties in effectively matching the distribution. (ii) **Distribution deviation with random noise.** Besides, these methods generate distilled images from the initial noise, which inject unpredictable randomness to latents, potentially violating the structural and representative information gathered in the VAE space with potential distribution deviation. (iii) **Separate sampling.** Moreover, the distilled data are individually matched to parts of the entire VAE distribution without considering the

---

[1]Northeastern University [2]The Ohio State University. Correspondence to: Pu Zhao <p.zhao@northeastern.edu>, Xue Lin <xue.lin@northeastern.edu>.

*Proceedings of the 42$^{st}$ International Conference on Machine Learning*, Vancouver, Canada. PMLR 267, 2025. Copyright 2025 by the author(s).

---

[1]Normality refers to the degree of the latent space data conforms to a normal distribution. A higher degree of normality indicates closer conformity.

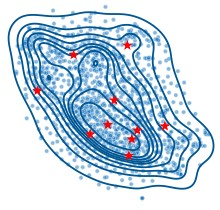 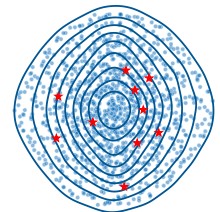

(a) Low-normality (VAE)    (b) High-normality(mapped)

*Figure 1.* **t-SNE visualization of the low-normality VAE space and high-normality noise space for class "Goldfish".** The blue contour lines are the probability density curves of the distribution using kernel density estimation, highlighting the structure and concentration of the latents (blue dots). ★ in (b) marks the 10 representative latents generated in the noise space, corresponding to ★ in (a) after DDIM sampling, which preserves the structure of VAE space and concentrates in high-density regions.

overall distribution of the distilled dataset, which may lead to an incomplete representation of the desired distribution.

Inspired by these insights, we propose a novel framework, Taming **D**iffusion for **D**ataset **D**istillation with **H**igh **R**epresentativeness (**D³HR**). We reveal an optimal strategy for modeling the VAE latent domain by mapping it to a simpler distribution domain while preserving fundamental information and ensuring structural consistency. Toward this end, we adopt the deterministic DDIM inversion to map latents from the VAE space to the noise space with higher normality, so that a Gaussian distribution can accurately match the noised latent distribution, as illustrated in Figure 1b. Then we sample a set of latents from the estimated Gaussian distribution and convert the latents back to VAE space by DDIM sampling for generating the distilled dataset. To ensure the distribution matching between the distilled dataset and the full dataset, we incorporate a new constraint with our group sampling method, which tights the similarity between the distribution of the generated latents and that of the original latents.

Our contributions can be summarized as follows:

- We propose D³HR, a dataset distillation framework leveraging DDIM inversion to map the original data distribution to a Gaussian distribution, which makes it easier for distribution matching (better normality) with less randomness (bijective DDIM mapping to preserve structural information), effectively addressing inaccurate distribution matching and distribution deviation in previous diffusion-based methods.
- We propose a novel sampling scheme to efficiently align the distributions of generated distilled latents and original latents from the full dataset, overcoming the shortcoming of individual sampling.
- We conduct extensive experiments and demonstrate that D³HR significantly outperforms state-of-the-art

dataset distillation methods, with detailed ablation studies and analysis.

## 2. Related Work

**Dataset Distillation**. Dataset distillation (Wang et al., 2018) is designed to improve training efficiency by generating a small dataset to replace the original large dataset for training. Previous methods (Zhao et al., 2020; Cazenavette et al., 2022) employ a bi-level optimization process, which updates the model parameters and the distilled dataset simultaneously. According to the different matching strategies in the distillation, the methods can be classified into: gradient matching (Zhao et al., 2020; Zhao & Bilen, 2021; Lee et al., 2022), distribution matching (Wang et al., 2022; Zhao & Bilen, 2023; Zhao et al., 2023; Zhang et al., 2023; Deng et al., 2024), and trajectory matching (Cazenavette et al., 2022; Cui et al., 2023; Du et al., 2023; 2024a).

Recent methods demonstrate that the bilevel optimization approach is time-consuming and does not scale well for large datasets (Loo et al.; Sun et al., 2024; Yin et al., 2023; Shao et al., 2024; Yu et al., 2024). To address this, more efficient dataset distillation methods are developed. SRe²L (Yin et al., 2023) and DWA (Du et al., 2024b) use the BN layers of the pre-trained teacher model as supervisory information and synthesize each distilled image individually to accelerate the generation. RDED (Sun et al., 2024) emphasizes the importance of image realism and generates images by stitching patches selected by the teacher model. However, these methods rely on BN layers or the data distribution of other layers from the teacher, which limits their applicability to different model architectures.

Unlike the above methods, the diffusion-based methods leverage the powerful generative capabilities of diffusion models, eliminating the dependency on teacher models. Specifically, D⁴M (Su et al., 2024) generates the distilled dataset by clustering the latents of the full dataset in the VAE latent space. Gu et al. (2024) reconstruct the training loss of diffusion model in the VAE space, aiming to generate images representing the full dataset. However, it requires fine-tuning multiple models for large datasets. The above methods struggle to accurately capture representative latents that match the full dataset distribution.

**Diffusion Models**. Diffusion models target approximating the data distribution $q_\theta(x_\theta)$ with a learned model distribution $p_\theta(x_\theta)$. Denoising diffusion probabilistic models (DDPMs) (Ho et al., 2020) optimizes a variational lower bound based on the Markov Chain, allowing models to generate high-quality samples. DDIM (Song et al., 2020) built on DDPMs provides a more efficient and deterministic sampling method by removing the randomness in each reverse step, thus offering a faster and more controlled generation.

Due to the deterministic approach, DDIM sampling can be inverted, allowing users to map their generated or real image back to its corresponding noise representation. Previous works (Rombach et al., 2022) rely on the UNet architecture, and recent works (Peebles & Xie, 2023; Esser et al., 2024; Shen et al., 2025) demonstrate the superior performance of Diffusion Transformer (DiT) in image generation by introducing Transformer architecture to diffusion models.

## 3. Background, Formulation, and Motivation

### 3.1. Preliminaries on Diffusion Models

*Diffusion Models* (Ho et al., 2020) are proposed to generate high-quality images by transforming the random Gaussian noise $x_t$ into the image $x_0$ through a discrete Markov chain. Among them, Latent Diffusion Model (Rombach et al., 2022) enhances efficiency by leveraging VAE to compress the pixel space $\mathcal{X}_0$ into the latent space $\mathcal{Z}_0 : z_0 = E(x_0)$, and decoding the latents back to images at the end of diffusion backward process: $x_0 = D(z_0)$. During training, the forward process adds random noise to the initial latent $z_0$:

$$z_t = \sqrt{\alpha_t} z_0 + \sqrt{1 - \alpha_t}\epsilon, \quad \text{with} \quad \epsilon \sim \mathcal{N}(0, \mathbf{I}), \quad (1)$$

where $\alpha_t$ is a hyper-parameter and $z_t$ represents the noise at timestep $t$. During inference, the backward process iteratively removes noise in $z_t$ through schedulers to get $z_0$.

*Denoising Diffusion Implicit Models* (DDIM) (Song et al., 2020) introduces a deterministic sampling method by configuring the variance of the distribution at each step, enabling a one-to-one mapping from $z_t$ to $z_0$, as expressed below:

$$z_{t-1} = \sqrt{\frac{\alpha_{t-1}}{\alpha_t}} z_t + \\ \sqrt{\alpha_{t-1}} \left( \sqrt{\frac{1}{\alpha_{t-1}} - 1} - \sqrt{\frac{1}{\alpha_t} - 1} \right) \varepsilon_\theta(z_t, t, \mathcal{C}).$$

$$(2)$$

where $\varepsilon_\theta(z_t, t, \mathcal{C})$ is a function with trainable parameters $\theta$, $\mathcal{C}$ denotes the class condition.

### 3.2. Problem Formulation for Dataset Distillation

Dataset distillation aims to synthesis a small distilled dataset $\mathcal{S} = \{\hat{\mathbf{x}}_i, \hat{y}_i\}_{i=1}^{N_\mathcal{S}}$ to replace the full dataset $\mathcal{F} = \{\mathbf{x}_i, y_i\}_{i=1}^{N_\mathcal{F}}$ for training, where $N_\mathcal{S} \ll N_\mathcal{F}$, $\hat{\mathbf{x}}_i$ & $\mathbf{x}_i$ are images, and $\hat{y}_i$ & $y_i$ are labels. To maintain high performance with a smaller $\mathcal{S}$, Algorithm $\mathcal{A}$ is proposed to address the problem of generating $\mathcal{S}$ from $\mathcal{F}$: $\mathcal{S} = \mathcal{A}(\mathcal{F}) \mid N_\mathcal{S} \ll N_\mathcal{F}$. It is expected that training a model on $\mathcal{S}$ achieves a comparable performance to training on $\mathcal{F}$, assuming that $\mathcal{S}$ encapsulates substantial information from the original $\mathcal{F}$.

### 3.3. Motivation

Existing dataset distillation works have certain limitations as discussed below, which motivate us to develop our D³HR.

**Scalability to multiple model architectures**. In some dataset distillation works (Yin et al., 2023; Loo et al.), a teacher model $T$ trained by $\mathcal{F}$ is used to guide the distillation process: $\mathcal{S} = \mathcal{A}(\mathcal{F}, T)$. It relies on a specific model architecture. A new model architecture requires distilling another dataset with the new teacher architecture expensively trained on the full dataset, making it hard to scale.

To address this problem, recent works (Gu et al., 2024; Su et al., 2024) leverage the powerful generative capabilities of diffusion models $G$ (Azizi et al., 2023) to produce $\mathcal{S}$ without the guidance from a specific teacher model $T$: $\mathcal{S} = \mathcal{A}(\mathcal{F}, G)$. In this setup, once $\mathcal{S}$ is generated with just a **one-time cost**, it can be applied to various model architectures (*e.g.*, ResNet, EfficientNet, and VGG), instead of generating multiple $\mathcal{S}$ for each model architecture.

**Inaccurate distribution matching in the latent space**. There are still some limitations for the current diffusion-based data distillation methods. Specifically, the original images of the class $\mathcal{C}$ are converted to latents $\mathcal{Z}_{0,\mathcal{C}} = \left\{ z_0^i \mid \mathcal{C} \right\}_{i=1}^{N_{\mathcal{F},\mathcal{C}}}$ through the VAE, and the methods propose to find and describe the data distributions in the latent space, for further synthesis of $n$ latents for $\mathcal{C}$ (i.e., $n$ images-per-class (IPC)), hoping that $n$ latents can match the real data distributions in the latent space. For example, Su et al. (2024) apply K-means to cluster $\mathcal{Z}_{0,\mathcal{C}}$ into $n$ groups to get the centroids for better synthesis of $n$ latents. Gu et al. (2024) minimize the cosine similarity between each synthesized latent and a random subset $\mathcal{Z}_s$, where $\mathcal{Z}_s \subset \mathcal{Z}_{0,\mathcal{C}}$.

However, the distribution of the original dataset in the latent space is too complex to accurately describe or match. It is a multi-component Gaussian mixture distribution as below.

**Lemma 3.1.** *Each latent in VAE latent space is randomly sampled from a distinct component of a multi-component Gaussian mixture distribution.*

The proof is shown in Appendix A.1. $\mathcal{Z}_{0,\mathcal{C}}$ is a multi-component Gaussian mixture distribution and hard-to-fit, with examples shown in Figure 1a.

Therefore, it is challenging to generate representative latents in the complex $\mathcal{Z}_{0,\mathcal{C}}$. The K-means algorithm (Su et al., 2024) assumes that each cluster is spherical for easier clustering $\mathcal{Z}_{0,\mathcal{C}}$, which does not align with the more complex distribution of $\mathcal{Z}_{0,\mathcal{C}}$. The cosine similarity (Gu et al., 2024) focuses on the direction to a subset of the distribution, and ignores the distance, making it less effective to capture the differences in probability density within $\mathcal{Z}_{0,\mathcal{C}}$.

**Distribution deviation with random noise**. Current meth-

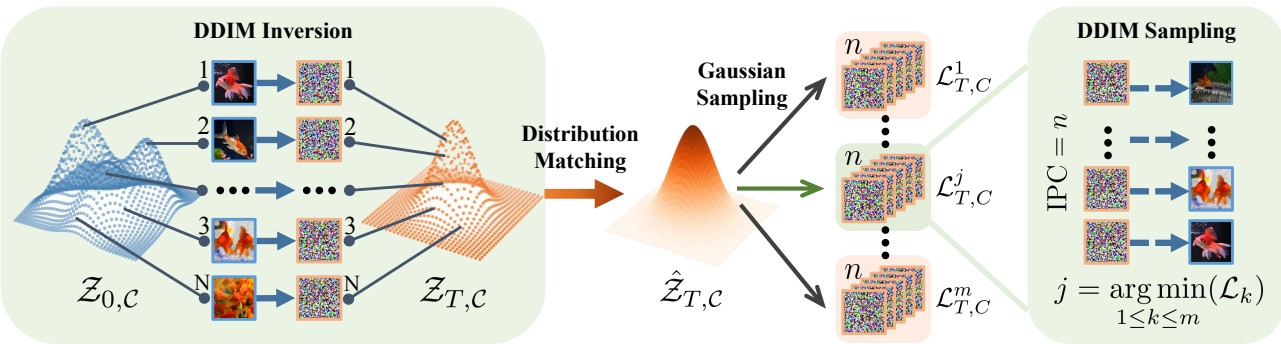

*Figure 2.* **Illustration of proposed D³HR framework.** With the latents from the VAE, DDIM inversion is applied to map the latent embeddings to a Gaussian domain with better normality, which can further be matched to a Gaussian distribution. Then, we follow Section 4.4 to sample representative latents based on different IPC requirements, and generate true images through DDIM sampling.

ods generate images from the initial noise. They obtain the noise either by adding random noise to the representative latents in the VAE space via Equation (1) (Su et al., 2024), or by generating it randomly, with the representative latents obtained by multiple fine-tuned diffusion models (Gu et al., 2024). However, due to the inherent randomness of initial noise, the distribution of denoised latents may deviate or shift from that of the sampled latents before adding noise, as shown in Appendix B-Figure A1, leading to potential accuracy loss after training.

**Separate sampling**. Additionally, current diffusion-based works generate representative latents without considering their overall distribution. Due to the limited number of samples, although each latent may be matched to parts of the desired distribution $\mathcal{Z}_{0,\mathcal{C}}$, the overall distribution of the sampled $n$ latents may not closely align with $\mathcal{Z}_{0,\mathcal{C}}$, resulting in an under-represented distilled dataset.

## 4. Methodology for Dataset Distillation

Motivated by the limitations, we present D³HR, a diffusion-based dataset distillation framework via domain mapping.

### 4.1. Framework Overview

Our framework has three stages: domain mapping, distribution matching, and group sampling. As shown in Figure 2, we first convert images to latents through VAE and map the VAE latent domain to the noise domain via DDIM inversion. Next, with distribution matching, we use a Gaussian distribution to match the mapped noised latent distribution from the previous stage. Finally, in the group sampling stage, we sample a number of latents from the Gaussian distribution and convert the latents back to images to generate the distilled dataset. The algorithm is shown in Algorithm 1 with details introduced in the following subsections.

### 4.2. Domain Mapping

It is essential that the small distilled dataset can sufficiently represent the original large data. Previous works (Gu et al., 2024; Su et al., 2024) use the distribution in VAE latent space $\mathcal{Z}_{0,\mathcal{C}} = \left\{ z_0^i \mid \mathcal{C} \right\}_{i=1}^{N}$ to distill the information of class $\mathcal{C}$ from the original dataset. As discussed in Section 3.3, it is challenging to generate $n$ latents to match the VAE latent space $\mathcal{Z}_{0,\mathcal{C}}$. To make it easier for distribution matching, we further map the distribution in $\mathcal{Z}_{0,\mathcal{C}}$ to a high-normality Gaussian space $\mathcal{Z}_{T,\mathcal{C}}$ by *DDIM inversion* (Dhariwal & Nichol, 2021) for each class $\mathcal{C}$, as illustrated in Figure 2. Specifically, for each latent $z_0$ in class $\mathcal{C}$, we perform a few DDIM inversion steps as follows,

$$
z_{t+1} = \sqrt{\frac{\alpha_{t+1}}{\alpha_t}} z_t +
$$
$$
\alpha_{t+1} \left( \sqrt{\frac{1}{\alpha_{t+1}} - 1} - \sqrt{\frac{1}{\alpha_t} - 1} \right) \varepsilon_\theta(z_t, t, \mathcal{C}).
\tag{3}
$$

*DDIM inversion* is based on the assumption that the ODE process of DDIM sampling can be inverted in a few steps. There are other choices for domain mapping such as directly adding random Gaussian noise through the forward of Denoising Diffusion Probabilistic Models (DDPM) in Equation (1). However, DDPM inherently introduces randomness in the mapping, leading to difficulties for accurately describing the distribution with potential distribution shift or information loss, as presented in Figure 3a. Different from DDPM, DDIM inversion with Equation (3) offers the key prerequisites for obtaining a representative subset in $\mathcal{Z}_{T,\mathcal{C}}$: (i) **Information Preservation:** The mapping between two domains is bijective because of the deterministic process, where each element in $\mathcal{Z}_{T,\mathcal{C}}$ directly corresponds to an element in $\mathcal{Z}_{0,\mathcal{C}}$ and vice versa, which avoids the loss of key features during the mapping process. (ii) **Structural Consistency:** The latents in $\mathcal{Z}_{T,\mathcal{C}}$ can retain the structural

**Algorithm 1** D³HR Algorithm

---

**Input:** Full dataset $\mathcal{F} = (X, Y)$, VAE encoder-decoder $E, D$, Inverse steps $T$, IPC number $n$
**Output:** Distilled dataset $\mathcal{S}$
Initialize $\mathcal{S} = \emptyset$
**for each** $class\ \mathcal{C}$ **do**
    $y_{\mathcal{C}} \leftarrow label\ of\ class\ \mathcal{C}$
    $\mathcal{Z}_{0,\mathcal{C}} = E(X|C)$
    Initialize $\mathcal{Z}_{T,\mathcal{C}} = \emptyset$
    **for each** $z_0 \in \mathcal{Z}_{0,\mathcal{C}}$ **do**
        Update $z_0$ to $z_T$ via Equation (3) with $T$ steps
                                ▷ DDIM Inversion
        $\mathcal{Z}_{T,\mathcal{C}} \leftarrow \mathcal{Z}_{T,\mathcal{C}} \cup \{z_T\}$
    **end for**
    $\mu_{T,\mathcal{C}} \leftarrow mean\ of\ \mathcal{Z}_{T,\mathcal{C}}$
    $\sigma^2_{T,\mathcal{C}} \leftarrow variance\ of\ \mathcal{Z}_{T,\mathcal{C}}$
    $\hat{\mathcal{Z}}_{T,\mathcal{C}} \leftarrow \mathcal{N}(\mu_{T,\mathcal{C}}, \sigma^2_{T,\mathcal{C}})$
    Initialize $\mathcal{L}_{T,\mathcal{C}} = \emptyset$            ▷ Group Sampling
    **for** $k \leftarrow 1$ to $m$ **do**
        Gaussian Sample $n$ latents to get $\mathcal{R}^k = \{\hat{z}^i_{T,\mathcal{C}}\}^n_{i=1}$
        Calculate $\mathcal{L}^k_{T,\mathcal{C}}$ via Equation (5) for $\mathcal{R}^k$
        $\mathcal{L}_{T,\mathcal{C}} = \mathcal{L}_{T,\mathcal{C}} \cup \{\mathcal{L}^k_{T,\mathcal{C}}\}$
    **end for**
    $j = \underset{1 \le k \le m}{\arg\min}(\mathcal{L}^k_{T,\mathcal{C}})$
    **for each** $\hat{z}_T \in \mathcal{R}^j$ **do**
        Update $\hat{z}_T$ to $\hat{z}_0$ via Equation (2) in $T$ steps
                               ▷ DDIM Sampling
        $\mathcal{S} \leftarrow \mathcal{S} \cup \{D(\hat{z}_0), y_{\mathcal{C}}\}$
    **end for**
**end for**
**return** $\mathcal{S}$

---

information of $\mathcal{Z}_{0,\mathcal{C}}$, ensuring the structural consistency and distribution alignment.

### 4.3. Distribution Matching

With DDIM inversion, we can obtain a discrete distribution $\mathcal{Z}_{T,\mathcal{C}} = \left\{ z^i_T \mid \mathcal{C} \right\}^N_{i=1}$. We have the following lemma.

**Lemma 4.1.** *For DDIM inversion with $T$ steps, with sufficiently large $T$, $\mathcal{Z}_{T,\mathcal{C}}$ can be approximated as a Gaussian distribution.*

The proof is shown in Appendix A.2. Since the latents in $\mathcal{Z}_{T,\mathcal{C}}$ can be interpreted as independently and identically Gaussian distributed (i.i.d.) latents, we approximate $\mathcal{Z}_{T,\mathcal{C}}$ with a Gaussian distribution $\hat{\mathcal{Z}}_{T,\mathcal{C}}$ based on the law of large numbers (Hsu & Robbins, 1947). To obtain the statistical properties of $\hat{\mathcal{Z}}_{T,\mathcal{C}}$, we compute the the mean $\mu_{T,\mathcal{C}}$ and variance $\sigma^2_{T,\mathcal{C}}$ from $\mathcal{Z}_{T,\mathcal{C}}$ as the mean and variance of $\hat{\mathcal{Z}}_{T,\mathcal{C}}$.

Thus, we can obtain the Gaussian distribution $\hat{\mathcal{Z}}_{T,\mathcal{C}} \sim \mathcal{N}(\mu_{T,\mathcal{C}}, \sigma^2_{T,\mathcal{C}})$ in the latent space to represent the class $\mathcal{C}$. Since the dimensions in the noise space of DDIM inversion are independent (Song et al., 2020), the probability density

function (PDF) of $\hat{\mathcal{Z}}_{T,\mathcal{C}}$ can be expressed as the following,

$$f(\hat{\mathbf{z}}_{T,\mathcal{C}}) = \prod_{i=1}^{d} \frac{1}{\sqrt{2\pi(\sigma^i_{T,\mathcal{C}})^2}} \exp\left(-\frac{(\hat{z}^i_{T,\mathcal{C}} - \mu^i_{T,\mathcal{C}})^2}{2(\sigma^i_{T,\mathcal{C}})^2}\right), \quad (4)$$

where $\mu^i_{T,\mathcal{C}}$ and $\sigma^i_{T,\mathcal{C}}$ denote the mean and standard deviation for each dimension of $\mu_{T,\mathcal{C}}$ and $\sigma_{T,\mathcal{C}}$, respectively.

### 4.4. Group Sampling

With the Gaussian distribution $\hat{\mathcal{Z}}_{T,\mathcal{C}}$ for the class $\mathcal{C}$, next we generate the distilled dataset. Specifically, we generate $n$ representative latents for each class $\mathcal{C}$ by capturing the statistic characteristics of $\hat{\mathcal{Z}}_{T,\mathcal{C}}$. Using the Ziggurat algorithm (Marsaglia & Tsang, 2000), we can randomly sample $n$ i.i.d. latents probabilistically following $\hat{\mathcal{Z}}_{T,\mathcal{C}}$ with $f(\hat{\mathbf{z}}_{T,\mathcal{C}})$, which together form a subset. However, although each latent is sampled from $\hat{\mathcal{Z}}_{T,\mathcal{C}}$, due to the limited number of samples ($n$) for distillation, the overall distribution of the subset with $n$ samples may still deviate from the desired $\hat{\mathcal{Z}}_{T,\mathcal{C}}$, leading to certain performance degradation.

To address this problem and further improve the quality of the distilled dataset, we propose to sample numerous random subsets (each subset consists of $n$ i.i.d. latents) and efficiently search for the most representative subset as the final subset. Specifically, by repeating the above sampling with the Ziggurat algorithm multiple times, $m$ random subsets are available with varying statistical distributions. To search for the most representative subset, we further propose an efficient algorithm that selects the subset statistically closest to $\hat{\mathcal{Z}}_{T,\mathcal{C}}$ among $m$ random subsets, thereby improving the matching **effectiveness** and **stability**.

To select the final subset, we design a statistic evaluation metric $\mathcal{L}_{T,\mathcal{C}}$ to measure the distance between the subset distribution and $\hat{\mathcal{Z}}_{T,\mathcal{C}}$, and select the subset with the minimal value. Specifically, $\mathcal{L}_{T,\mathcal{C}}$ computes the differences in the key statistics between two data distributions, including the mean, standard deviation, and skewness. $\mathcal{L}_{T,\mathcal{C}}$ can be expressed as the weighted sum of the three,

$$\mathcal{L}_{T,\mathcal{C}} = \lambda_\mu \cdot \mathcal{L}_{\mu,T,\mathcal{C}} + \lambda_\sigma \cdot \mathcal{L}_{\sigma,T,\mathcal{C}} + \lambda_{\gamma_1} \cdot \mathcal{L}_{\gamma_1,T,\mathcal{C}}, \quad (5)$$

where $\mathcal{L}_{\mu,T,\mathcal{C}}, \mathcal{L}_{\sigma,T,\mathcal{C}}$ and $\mathcal{L}_{\gamma_1,T,\mathcal{C}}$ denotes the evaluation of $\mu_{T,\mathcal{C}}, \sigma_{T,\mathcal{C}}$, and $\gamma_{1,T,\mathcal{C}}$, respectively, with the formulation below,

$$\mathcal{L}_{\mu,T,\mathcal{C}} = (\bar{z}_{T,\mathcal{C}} - \mu_{T,\mathcal{C}})^2 = (\tfrac{1}{n}\sum_{i=1}^{n} \hat{z}^i_{T,\mathcal{C}} - \mu_{T,\mathcal{C}})^2, \quad (6)$$

$$\mathcal{L}_{\sigma,T,\mathcal{C}} = \left(\sqrt{\tfrac{1}{n}\sum_{i=1}^{n}(\hat{z}^i_{T,\mathcal{C}} - \mu_{T,\mathcal{C}})^2} - \sigma_{T,\mathcal{C}}\right)^2, \quad (7)$$

$$\mathcal{L}_{\gamma_1,T,\mathcal{C}} = \left(\tfrac{n}{(n-1)(n-2)}\sum_{i=1}^{n}\left(\tfrac{\hat{z}^i_{T,\mathcal{C}} - \mu_{T,\mathcal{C}}}{\sigma_{T,\mathcal{C}}}\right)^3 - \gamma_{1,T,\mathcal{C}}\right)^2, \quad (8)$$

where $\mu_{T,\mathcal{C}}, \sigma_{T,\mathcal{C}}$ and $\gamma_{1,T,\mathcal{C}}$ are the mean, standard deviation and skewness of $\hat{\mathcal{Z}}_{T,\mathcal{C}}$. $\hat{z}^i_{T,\mathcal{C}}$ indicates the $i^{th}$ element

| Dataset | IPC | ResNet-18 | | | | | ResNet-101 | | | | |
|---|---|---|---|---|---|---|---|---|---|---|---|
| | | SRe²L | DWA | D⁴M | RDED | Ours | SRe²L | DWA | D⁴M | RDED | Ours |
| CIFAR-10 | 10 | $29.3 \pm 0.5$ | $32.6 \pm 0.4$ | 33.5 | $37.1 \pm 0.3$ | $\mathbf{41.3 \pm 0.1}$ | $24.3 \pm 0.6$ | $25.2 \pm 0.2$ | 29.4 | $33.7 \pm 0.3$ | $\mathbf{35.8 \pm 0.6}$ |
| | 50 | $45.0 \pm 0.7$ | $53.1 \pm 0.3$ | 62.9 | $62.1 \pm 0.1$ | $\mathbf{70.8 \pm 0.5}$ | $34.9 \pm 0.1$ | $48.2 \pm 0.4$ | 54.4 | $51.6 \pm 0.4$ | $\mathbf{63.9 \pm 0.4}$ |
| CIFAR-100 | 10 | $31.6 \pm 0.5$ | $39.6 \pm 0.6$ | 38.1 | $42.6 \pm 0.2$ | $\mathbf{49.4 \pm 0.2}$ | $30.7 \pm 0.3$ | $35.9 \pm 0.5$ | 33.3 | $41.1 \pm 0.2$ | $\mathbf{46.0 \pm 0.5}$ |
| | 50 | $52.2 \pm 0.3$ | $60.9 \pm 0.5$ | 63.2 | $62.6 \pm 0.1$ | $\mathbf{65.7 \pm 0.3}$ | $56.9 \pm 0.1$ | $58.9 \pm 0.6$ | 64.7 | $63.4 \pm 0.3$ | $\mathbf{66.6 \pm 0.2}$ |
| Tiny-ImageNet | 10 | $16.1 \pm 0.2$ | $40.1 \pm 0.3$ | 34.1 | $41.9 \pm 0.2$ | $\mathbf{44.4 \pm 0.1}$ | $14.6 \pm 1.1$ | $38.5 \pm 0.7$ | 33.4 | $22.9 \pm 3.3$ | $\mathbf{43.2 \pm 0.5}$ |
| | 50 | $41.4 \pm 0.4$ | $52.8 \pm 0.2$ | 46.2 | $\mathbf{58.2 \pm 0.1}$ | $56.9 \pm 0.2$ | $42.5 \pm 0.2$ | $54.7 \pm 0.3$ | 51.0 | $41.2 \pm 0.4$ | $\mathbf{59.4 \pm 0.1}$ |
| | 100 | $49.7 \pm 0.3$ | $56.0 \pm 0.2$ | 51.4 | - | $\mathbf{59.3 \pm 0.1}$ | $51.5 \pm 0.3$ | $57.4 \pm 0.3$ | 55.3 | - | $\mathbf{61.4 \pm 0.1}$ |
| ImageNet-1K | 10 | $21.3 \pm 0.6$ | $37.9 \pm 0.2$ | 27.9 | $42.0 \pm 0.1$ | $\mathbf{44.3 \pm 0.3}$ | $30.9 \pm 0.1$ | $46.9 \pm 0.4$ | 34.2 | $48.3 \pm 1.0$ | $\mathbf{52.1 \pm 0.4}$ |
| | 50 | $46.8 \pm 0.2$ | $55.2 \pm 0.2$ | 55.2 | $56.5 \pm 0.1$ | $\mathbf{59.4 \pm 0.1}$ | $60.8 \pm 0.5$ | $63.3 \pm 0.7$ | 63.4 | $61.2 \pm 0.4$ | $\mathbf{66.1 \pm 0.1}$ |
| | 100 | $52.8 \pm 0.3$ | $59.2 \pm 0.3$ | 59.3 | - | $\mathbf{62.5 \pm 0.0}$ | $62.8 \pm 0.2$ | $66.7 \pm 0.2$ | 66.5 | - | $\mathbf{68.1 \pm 0.0}$ |

*Table 1.* **Comparison of Top-1 accuracy across four datasets with other methods.** Following SRe²L (Yin et al., 2023), all methods use ResNet-18 as teacher model, and train separately on ResNet-18 and ResNet-101 with the soft label. As D⁴M (Su et al., 2024) and our D³HR do not need teacher models, we just use the soft label as supervision. For IPC = 100, some results are unavailable with '-' due to their inappropriate default parameter settings.

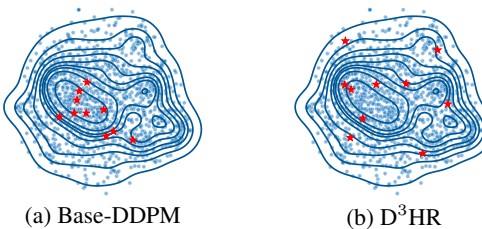

(a) Base-DDPM  (b) D³HR

*Figure 3.* **t-SNE visualization of the representative latents in VAE space generated by Equation (1) and D³HR for class "Goldfish".** It can be observed that the latents accurately represent the VAE distribution in D³HR.

of the random subset. As Gaussian distribution is perfectly symmetric, $\gamma_{1,T,\mathcal{C}} = 0$.

With the above evaluation metric function, for the $k^{th}$ subset, we can compute its metric $\mathcal{L}_{T,\mathcal{C}}^{k}$. And we select the $j^{th}$ subset with the smallest $\mathcal{L}_{T,\mathcal{C}}$ as the most representative subset:

$$j = \underset{1 \le k \le m}{\arg\min}(\mathcal{L}_{T,\mathcal{C}}^{k}). \tag{9}$$

After we select the most representative subset, they are converted to real images through Equation (2) with DDIM and a VAE decoder.

### 4.5. Advantages of D³HR

Compared to other methods, we highlight our approach has the following notable advantages: (1) We present a domain mapping method to map VAE latent domain into Gaussian domain with **better normality**, leading to more **accurate** and **efficient** distribution matching, as illustrated in Figure 1b. (2) We ensure that **the entire subset is representative** closely aligned with the desired distribution, rather than just sampling individual elements without considering their overall distribution. (3) Our sampling algorithm

is **both time- and space-efficient**. The sampling is performed based on the statistic parameters and we can sample different subsets in parallel on the GPU. Our sampling implementation only takes 2.6 seconds per class on a single RTX A6000 with $m = 1e6$ for ImageNet-1K, IPC = 10. Additional runtime results under different settings are provided in Appendix B-Tables A2 to A4. Moreover, instead of storing representative latents, we can generate the distilled image directly from statistical parameters, which are space-efficient as shown in Appendix B-Figure A2.

## 5. Main Results

### 5.1. Experimental Details

**Datasets**. Experiments are conducted on both small-scale and large-scale datasets. For small-scale datasets, we use CIFAR-10 and CIFAR-100 (Krizhevsky et al., 2009) with $32 \times 32$ resolution. For large-scale datasets, we use Tiny-ImageNet (Le & Yang, 2015) with 200 classes (500 images per class, $64 \times 64$ size) and ImageNet-1K (Deng et al., 2009) with $1,000$ classes (1M images, $224 \times 224$ resolution).

**Network architectures**. To validate the applicability of our method across different architectures, we adopt ResNet-18/ResNet-101 (He et al., 2016), MobileNet-V2 (Sandler et al., 2018), and VGG-11 (Simonyan & Zisserman, 2014) as backbones, following prior works (Du et al., 2024b; Yin et al., 2023; Sun et al., 2024).

**Baselines**. We compare D³HR with four state-of-the-art methods: SRe²L (Yin et al., 2023), DWA (Du et al., 2024b), D⁴M (Su et al., 2024), RDED (Sun et al., 2024), following the same evaluation configuration. We exclude comparison with Minimax (Gu et al., 2024) in the main Table 1, since their method focuses on handling small subsets of ImageNet-1K and requires training multiple diffusion mod-

| Student\Teacher | | ResNet-18 | MobileNet-V2 | VGG-11 |
|---|---|---|---|---|
| ResNet-18 | $SRe^2L$ | $21.3 \pm 0.6$ | $15.4 \pm 0.2$ | - |
| | RDED | $42.3 \pm 0.6$ | $40.4 \pm 0.1$ | $36.6 \pm 0.1$ |
| | Ours | $\mathbf{44.3 \pm 0.3}$ | $\mathbf{42.3 \pm 0.7}$ | $\mathbf{38.3 \pm 0.2}$ |
| MobileNet-V2 | $SRe^2L$ | $19.7 \pm 0.1$ | $10.2 \pm 2.6$ | - |
| | RDED | $34.4 \pm 0.2$ | $33.8 \pm 0.6$ | $28.7 \pm 0.2$ |
| | Ours | $\mathbf{43.4 \pm 0.3}$ | $\mathbf{46.4 \pm 0.2}$ | $\mathbf{37.8 \pm 0.4}$ |
| VGG-11 | $SRe^2L$ | $16.5 \pm 0.1$ | $10.6 \pm 0.1$ | - |
| | RDED | $22.7 \pm 0.1$ | $21.6 \pm 0.2$ | $23.5 \pm 0.3$ |
| | Ours | $\mathbf{25.7 \pm 0.4}$ | $\mathbf{24.8 \pm 0.4}$ | $\mathbf{28.1 \pm 0.1}$ |

*Table 2.* **Evaluating Top-1 accuracy for cross-architecture generalization on ImageNet-1K**, IPC = 10. As VGG-11 lacks BN layers, the results of $SRe^2L$ are not available with '-'.

| DM / Sampling Design | Acc. (%) |
|---|---|
| Base-DDPM | $37.3 \pm 0.7$ |
| Base-RS | $41.6 \pm 0.2$ |
| $\mathcal{L}_\mu$ | $42.6 \pm 0.1$ |
| $\mathcal{L}_\sigma$ | $42.3 \pm 0.2$ |
| $\mathcal{L}_{\gamma_1}$ | $42.4 \pm 0.1$ |
| $\mathcal{L}_\mu + \mathcal{L}_\sigma$ | $43.3 \pm 0.2$ |
| $\mathcal{L}_\mu + \mathcal{L}_{\gamma_1}$ | $43.0 \pm 0.1$ |
| $\mathcal{L}_\sigma + \mathcal{L}_{\gamma_1}$ | $42.5 \pm 0.2$ |
| $\mathcal{L}_\mu + \mathcal{L}_\sigma + \mathcal{L}_{\gamma_1}$ **(Ours)** | $\mathbf{44.3 \pm 0.3}$ |

*Table 3.* **Ablations of $D^3HR$ on ResNet-18 for ImageNet-1K with IPC = 10.** 'DM' indicates domain mapping, and 'RS' indicates random sampling.

els with expensive computation for large-scale scenarios. Instead, we report results under their setup in Section 6.4. **Implementation details**. We adopt the pre-trained Diffusion Transformer (DiT) and VAE from Peebles & Xie (2023) in our framework, originally trained on ImageNet-1K. We further adjust the conditioning labels of DiT, and fine-tune the pre-trained model with 400 epochs for each dataset (Tiny-ImageNet, CIFAR-10 and CIFAR-100) to adapt its generative capacity to the specific data distributions. For distillation, we employ 31 steps for DDIM inversion and sampling. During the validation, we follow other works (Yin et al., 2023; Du et al., 2024b; Su et al., 2024; Sun et al., 2024) to use soft-label of the teacher model as supervision for training. All experiments are conducted on Nvidia RTX A6000 GPUs or Nvidia A100 40GB GPUs.

### 5.2. Comparison with State-of-the-art Methods

As shown in Table 1, **$D^3HR$ demonstrates superior performance across all IPCs compared with baselines**.

**Large-scale datasets**. We first validate the practicality of $D^3HR$ on Tiny-ImageNet and ImageNet-1K at various IPCs. For Tiny-ImageNet, while RDED performs well on ResNet-18, its reliance on the teacher model leads to a significant drop on cross ResNet-101. In contrast, $D^3HR$ consistently delivers SOTA performance on both ResNet-18 and ResNet-101. Similarly, for ImageNet-1K, $D^3HR$ achieves higher performance across all IPCs compared with all baselines.

**Small-scale datasets**. For CIFAR-10 and CIFAR-100, $D^3HR$ surpasses all baselines. Notably, compared with the best-performing baseline RDED, on CIFAR-10, our method achieves significant improvements of 12.5% for ResNet-18 and 17.4% for ResNet-101 at 50 IPC.

In addition, we provide a comparison with other state-of-the-art methods that use validation settings different from ours. To ensure fairness, we evaluate our method under their settings, as shown in Tables A6 and A7, to demonstrate our superiority.

### 5.3. Cross-architecture Generalization

To further evaluate cross-model performance, on ImageNet-1K, we compare $D^3HR$ with the recent baseline $SRe^2L$ and RDED (SOTA performance on ImageNet-1K). For $D^3HR$, the teacher model refers to using its soft label for validation. As depicted in Table 2, we achieve a significant accuracy improvement across all cross-model evaluations. We highlight that with $D^3HR$, **a one-time cost is sufficient to achieve satisfactory results across various models**. Other baselines need to run their algorithms multiple times to generate multiple distilled datasets when the model architecture changes. The comparison results for more architectures are provided in Appendix B-Table A8.

## 6. Analysis

### 6.1. Ablation Studies

**Effectiveness of Domain Mapping by DDIM Inversion**. To demonstrate the effectiveness of domain mapping with DDIM inversion, we experiment with two configurations: (i) domain mapping with Equation (1) (denoted as Base-DDPM), and (ii) domain mapping with Equation (3) (denoted as Base-RS). Note that for both configurations, we perform distribution matching and individual Gaussian sampling (i.e., random sampling (RS)), while our proposed group sampling method in Section 4.4 is not applied.

As shown in Table 3, by comparing the results of Base-DDPM and Base-RS, we can observe that domain mapping with DDIM inversion outperforms that of DDPM with an accuracy improvement of 11.5%. As we discussed in Section 4.2, the added noise through DDPM causes structural information loss and domain shift for the mapping from $\mathcal{Z}_{0,\mathcal{C}}$ to $\mathcal{Z}_{T,\mathcal{C}}$, leading to difficulties to obtain a representative subset for $\mathcal{Z}_{0,\mathcal{C}}$. By contrast, the determinism of DDIM inversion resolves the issue with its information preservation and structural consistency, as illustrated in Figure 3b.

**Effectiveness of Group Sampling**. As shown in Table 3,

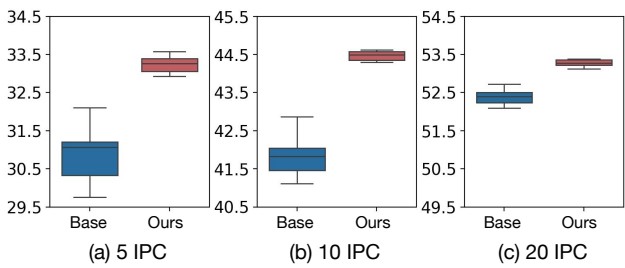

Figure 4. **Validation of the effectiveness and stability of group sampling on ImageNet-1K.** For our D³HR and Base-RS, we generate 3 distilled datasets at each IPC. Each dataset undergoes 3 rounds of validation, resulting in 9 data points per box plot.

by comparing our results with Base-RS, our proposed group sampling method can significantly outperform individual Gaussian sampling, demonstrating its effectiveness.

Furthermore, we conduct ablation studies with different combinations in Equation (5) to verify the contribution of each metric in the sampling schedule. As demonstrated in Table 3, each metric individually increases the accuracy, indicating that the subset selection effectively brings the distribution of the sampled representative subset closer to $\hat{\mathcal{Z}}_{T,\mathcal{C}}$. Combining all the metrics can lead to the best performance with the most representative subset.

Besides, we highlight that our group sampling method enhances the stability of the distilled dataset. As presented in Figure 4, we generate distilled datasets for the same dataset multiple times with both our D³HR and Base-RS. The results demonstrate that D³HR not only improves the accuracy but also enhances stability with smaller variance.

### 6.2. Analysis of Different Inversion Steps

In Lemma 4.1, as the number of inversion steps $t$ increase, $\hat{\mathcal{Z}}_{t,\mathcal{C}}$ gradually transits from a complex Gaussian mixture distribution to a standard Gaussian distribution, leading to a easier distribution matching. We provide feature visualizations to illustrate the changes as $t$ increases in Appendix B-Figure A3, validating Lemma 4.1. However, as $t$ increase, the added noise also increases, gradually diminishing the retention of original image structural information and reducing the reconstruction quality of DDIM inversion.

Specifically, there is a trade-off between maintaining the Gaussian assumption and preserving image structural information across different steps $t$. To assess the impact of different steps, we conduct experiments with varying $t$. As shown in Figure 5, the accuracy initially improves with increasing $t$, reaches a peak at $t = 31$, and then starts to decline. When $t$ is small (e.g., $t = 20$), the distribution is a mixture of Gaussians, and our distribution matching with a single Gaussian in Lemma 4.1 is not able to accurately describe the Gaussian mixtures, leading to certain perfor-

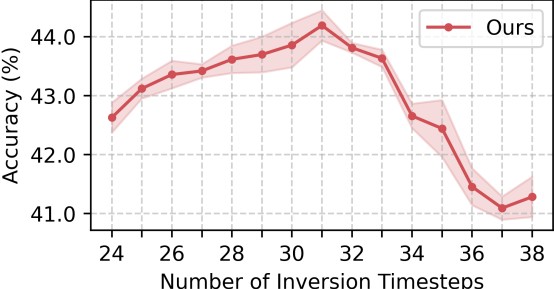

Figure 5. **The accuracy variation under different inversion timesteps for ImageNet-1K**, IPC $= 10$.

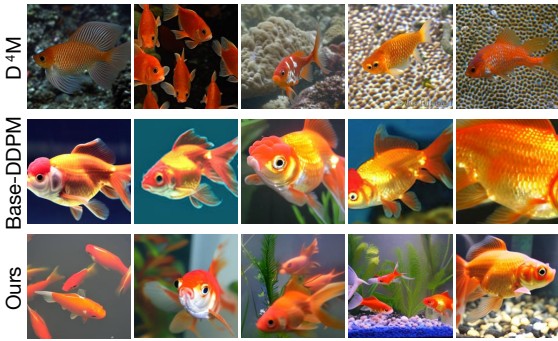

Figure 6. **Visualization of the distilled images for the class "gold-fish" on ImageNet-1K**, IPC $= 5$.

mance loss. When $t$ becomes large (e.g., $t = 40$), although our distribution matching can accurately represent the real distributions which becomes more normal, the real distributions suffer from more significantly structural information loss due to adding more noise, which in turn degrades the performance of DDIM inversion.

### 6.3. Image Visualization

We present the visualization results of distilled images generated by D4M, Base-DDPM, and our D³HR. Figure 6 shows that the images generated by D⁴M (Su et al., 2024) are simplistic and unrepresentative due to the incorrect retrieval of representative latents. For Base-DDPM with Equation (1), the noise space fails to generate representative latents due to randomness, resulting in outputs with overly simplistic structures, corresponding to Figure 3a. In contrast, the images generated by D³HR are both representative and diverse.

### 6.4. D³HR Outperforms Under Hard Labels

We give the comparison results with Minimax (Gu et al., 2024), which is the state-of-the-art method with hard labels under their main evaluation setting on ImageWoof (Howard, 2019). As demonstrated in Table 5, D³HR with 224 × 224 resolution outperforms Minimax (Gu et al., 2024) with 256 × 256 resolution.

| Dataset | IPC | ResNet-18 | | ResNet-101 | |
|---|---|---|---|---|---|
| | | D3S | Ours | D3S | Ours |
| ImageNet-1K | 10 | $39.1 \pm 0.3$ | $\mathbf{44.8 \pm 0.1}$ | $42.1 \pm 3.8$ | $\mathbf{52.8 \pm 0.6}$ |
| | 50 | $60.2 \pm 0.1$ | $\mathbf{60.2 \pm 0.0}$ | $65.3 \pm 0.5$ | $\mathbf{66.8 \pm 0.1}$ |

*Table 4.* **Comparison with D3S (Loo et al.) on ImageNet-1K.** Following the default setup of D3S with 5 pre-trained teacher models, our $D^3HR$ apply soft labels from these 5 teacher models.

| IPC | Model | Minimax | Ours |
|---|---|---|---|
| 10 | ResNet-18 | $37.6 \pm 0.9$ | $\mathbf{39.6 \pm 1.0}$ |
| | ResNetAP-10 | $39.2 \pm 1.3$ | $\mathbf{40.7 \pm 1.0}$ |
| 50 | ResNet-18 | $57.1 \pm 0.6$ | $\mathbf{57.6 \pm 0.4}$ |
| | ResNetAP-10 | $56.3 \pm 1.0$ | $\mathbf{59.3 \pm 0.4}$ |
| 100 | ResNet-18 | $65.7 \pm 0.4$ | $\mathbf{66.8 \pm 0.6}$ |
| | ResNetAP-10 | $64.5 \pm 0.2$ | $\mathbf{64.7 \pm 0.3}$ |

*Table 5.* **Comparison with Minimax (Gu et al., 2024) under hard labels across different models and IPCs.**

### 6.5. Improving Results with More Soft Labels

D3S (Loo et al.) proposes to simultaneously map the distribution of multiple teacher models. During evaluation, the soft labels from multiple teacher models are averaged to supervise the training of the student model. The drawback is the increased costs to train multiple different teachers.

Although $D^3HR$ does not rely on teacher models during the distillation, we find that using multiple soft labels during evaluation can further improve the performance. We present $D^3HR$ and D3S with 5 teacher models following their default setting on ImageNet-1K in Table 4. Since $D^3HR$ with one soft label outperforms D3S with five on TinyImageNet across various IPCs, we do not experiment with TinyImageNet. As shown in Table 1 and Table 4, leveraging 5 soft labels yields slight improvements over 1 soft labels across different IPCs and ours consistently outperforms D3S. For 10 IPC, $D^3HR$ with 1 soft label outperforms D3S with 5 soft labels on ImageNet-1K.

### 6.6. Robustness of $D^3HR$

As shown in Algorithm 1, $D^3HR$ involves inherent randomness: initializing different seeds for the random sampling of multiple subsets. For the diffusion-based $D^4M$ (Su et al., 2024), the randomness arises from two main sources: (i) Different seeds result in varying initial clusters in the K-means algorithm, and (ii) Different seeds generate varying initial noise by Equation (1) for the distilled dataset generation.

To validate the consistent superiority across various seeds of $D^3HR$, we conduct both our $D^3HR$ and $D^4M$ (Su et al., 2024) ten times with different seeds. As shown in Figure 7, our $D^3HR$ outperforms $D^4M$ (Su et al., 2024) by approxi-

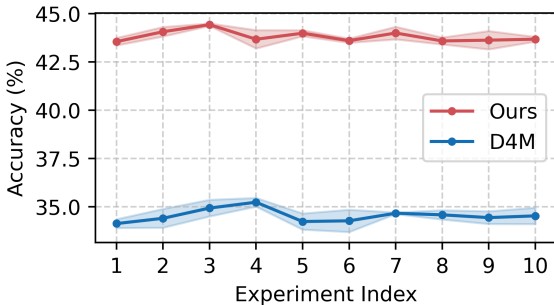

*Figure 7.* **The robustness of $D^3HR$ across different seeds on TinyImageNet.** IPC $= 10$. The "Experiment Index" represents ten different seeds selected at random.

mately $27.5\%$ across various seeds.

### 6.7. Storage Requirements Smaller than $D$

For $D^3HR$, all the data information is encompassed in $\hat{\mathcal{Z}}_T$. Therefore, we highlight that by storing only the statistical parameters (mean and variance) of $\hat{\mathcal{Z}}_T$ and the DiT pretrained weights of the diffusion model, we can preserve all the details of the distilled datasets under any IPCs. As shown in Appendix B-Figure A2, it further reduces the required storage space compared to $\mathcal{D}$. This approach is particularly effective for large datasets, resulting in a substantial reduction in storage requirements.

## 7. Conclusion

In this work, we thoroughly examine the challenges in existing diffusion-based dataset distillation methods, such as inaccurate distribution matching, distribution deviation with random noise, and the reliance on separate sampling. Based on this, we introduce a novel diffusion-based framework, $D^3HR$, to generate a highly representative distilled dataset. Our method achieves state-of-the-art validation performance on various datasets for different models.

## Impact Statement

This paper presents work whose goal is to advance the field of Machine Learning. There are many potential societal consequences of our work, none which we feel must be specifically highlighted here.

## Acknowledgments

This work is partially supported by the National Science Foundation under Award IIS-2310254. We would like to express our sincere gratitude to the reviewers for their invaluable feedback and constructive comments to improve the paper.

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

# Appendix

In the appendix, we include additional details that were omitted in the main text:

## A. Theoretical Analysis

**Lemma A.1.** *Each latent in VAE latent space is randomly sampled from a distinct component of a multi-component Gaussian mixture distribution; hence, the VAE latent is hard-to-fit.*

*Proof.* The Variational Autoencoder (VAE) (Kingma et al., 2019) learns the latent distribution $\mathcal{Z} = \{z_i | i = 1, ..., M\}$ of input images $\mathcal{X} = \{x_i | i = 1, ..., M\}$ through probabilistic modeling and is capable of sampling from this distribution to generate new images. In training, for each image $x_i$, VAE aims to maximize the marginal likelihood $p(x_i)$. The loss function can be defined by minimizing the negative evidence lower bound:

$$\mathcal{L}_i = -\mathbb{E}_{q(z_i|x_i)}\big[\log p(x_i|z_i)\big] + \text{KL}\big(q(z_i|x_i)\|p(z)\big). \tag{10}$$

The first term represents the reconstruction loss of $x_i$. The second term is the KL divergence loss, which is used to measure the difference between the latent distribution $q(z_i|x_i)$ produced by the encoder and the prior distribution $p(z) \sim \mathcal{N}(0, I)$. The output of the VAE encoder for $x_i$ is defined as a Gaussian distribution $q(z_i|x_i) \sim \mathcal{N}(\mu_i, \sigma_i^2)$, where $\mu_i$ and $\sigma_i^2$ are the mean and variance learned by the encoder. The latent $z_i$ is then obtained by randomly sampling from this Gaussian distribution. Therefore, for each class $\mathcal{C}$ containing $\mathbf{m}$ images, $\mathcal{Z}_\mathcal{C}$ is modeled as a discrete distribution, where the $\mathbf{m}$ latents are independently sampled from $\mathbf{m}$ distinct Gaussian distributions. The probability density function of the multi-component Gaussian mixture distribution can be expressed as:

$$p(z) = \sum_{i=1}^{\mathbf{m}} \alpha_i \cdot \mathcal{N}(x \mid \mu_i, \Sigma_i), \tag{11}$$

where $\mathcal{N}(x \mid \mu_i, \Sigma_i)$ denotes the $i$-th Gaussian distribution, with $\mu_i$ and $\Sigma_i$ representing its mean and covariance, respectively. The weights $\alpha_i$ satisfy the following condition: $\sum_{i=1}^{\mathbf{m}} \alpha_i = 1$. Since the position of each Gaussian component of this Gaussian mixture distribution is random, it results in a complex and hard-to-fit distribution.

**Lemma A.2.** *For DDIM inversion with $T'$ steps, when $t$ ($0 \le t \le T'$) is sufficiently large, $\hat{\mathcal{Z}}_{t,\mathcal{C}}$ can be approximated as a Gaussian distribution.*

*Proof.* In DDIM training, the forward process also follows Equation (1), which implies that $p(x_{T'}) \sim \mathcal{N}(0, I)$. During inference, the process from $x_T$ to $x_0$ is deterministic, following Equation (2). Since $\alpha_t$ is a pre-defined hyperparameter for each $t$, the process from $z_t$ to $z_{t-1}$ can be viewed as a linear transformation:

$$z_{t-1} = v_t \cdot z_t + w_t, \quad \text{where} \quad v_t = \sqrt{\frac{\alpha_{t-1}}{\alpha_t}}, \quad w_t = \sqrt{\alpha_{t-1}}\left(\sqrt{\frac{1}{\alpha_{t-1}} - 1} - \sqrt{\frac{1}{\alpha_t} - 1}\right)\varepsilon_\theta(z_t, t, \mathcal{C}). \tag{12}$$

For different sampled latent $z_t^i$, $v_t^i$ remains the same, while $w_t^i$ changes because the network $\varepsilon_\theta$ produces different outputs for each $z_t^i$.

We approach from the DDIM sampling, which is the reverse of the DDIM inversion, proceeding from $T'$ to 0. Specifically, consider the deterministic transformation from the set $\mathcal{Z}_{T',\mathcal{C}} = \{z_{T'}^i | i = 1, \ldots, \mathbf{m}\}$ to $\mathcal{Z}_{T'-1,\mathcal{C}}$ (each noise in $\mathcal{Z}_{T',\mathcal{C}}$ corresponds one-to-one with each latent in $\mathcal{Z}_{0,\mathcal{C}}$). Each latent in $\mathcal{Z}_{T'-1,\mathcal{C}}$ is a latent sampled from the Gaussian distribution:

$$z_{T'-1}^i \sim \mathcal{N}(w_{T'}^i, (v_{T'}^i)^2). \tag{13}$$

According to the law of large numbers (Hsu & Robbins, 1947), the latents in the discrete distribution $\mathcal{Z}_{T'-1,\mathcal{C}}$ can be interpreted as i.i.d. latents from continuous distribution $\hat{\mathcal{Z}}_{T'-1,\mathcal{C}}$. Thus, the $\hat{\mathcal{Z}}_{T'-1,\mathcal{C}}$ is a Gaussian mixture distribution.

Next, consider the transformation from $\mathcal{Z}_{T'-1,\mathcal{C}}$ to $\mathcal{Z}_{T'-2,\mathcal{C}}$. Each latent in $\mathcal{Z}_{T'-2,\mathcal{C}}$ is sampled from:

$$z^i_{T'-2} \sim \mathcal{N}(v_{T'-1} \cdot w^i_{T'} + w^i_{T'-1}, (v_{T'-1})^2 \times (v_{T'})^2). \tag{14}$$

Based on above iteration, the transformation from $\mathcal{Z}_{t+1,\mathcal{C}}$ to $\mathcal{Z}_{t,\mathcal{C}}$ can be generalized as follows:

$$z^i_t \sim \mathcal{N}\left( \sum_{k=t+1}^{T'} \left( \prod_{j=k+1}^{T'} v_{T'+t-j+1} \right) w^i_{T'+t-k+1}, \prod_{j=t+1}^{T'} (v_j)^2 \right), \tag{15}$$

Therefore, $\hat{\mathcal{Z}}_{t,\mathcal{C}}$ ($0 \leq t \leq T'-1$) is a Gaussian mixture distribution.

Now, in the DDIM sampling process, we have the recurrence relation for $\alpha_t$:

$$\alpha_t = \prod_{i=1}^{t} \alpha'_i, \quad \text{where} \quad \alpha'_i < 1 \& \alpha'_i \to 1, \tag{16}$$

this implies the following formulas:

$$|\alpha_t - \alpha_{t-1}| < \epsilon, \quad \text{where} \quad \epsilon \text{ is a small positive constant}, \tag{17}$$

$$\alpha_t \to 0 \quad \text{as} \quad t \to T'. \tag{18}$$

When $t$ is relatively large, on one hand, as indicated by Equations (12), (17) and (18), the effect of $\varepsilon_\theta(z_t, t, \mathcal{C})$ on $w_t$ becomes minimal; on the other hand, since the number of iterations is small, the change in the statistical properties of the original standard Gaussian distribution is not particularly noticeable. Hence, the distributions of the Gaussian components in $\hat{\mathcal{Z}}_{t,\mathcal{C}}$ are not significantly different, and $\hat{\mathcal{Z}}_{t,\mathcal{C}}$ can be well-approximated by a Gaussian distribution.

## B. More Implementation Details

For the group sampling, we set $m = 100,000$ for the ImageNet-1K, $m = 1,000,000$ for the TinyImageNet, and $m = 5,000,000$ for CIFAR-10, CIFAR-100. The hyper-parameters $\lambda_\mu, \lambda_\sigma, \lambda_{\gamma_1}$ are set to $1, 1, 0.5$, respectively. Regarding the computational cost of DDIM inversion, our method only requires approximately 4.5 hours on a single node with 8 A100 40G GPUs on ImageNet-1K.

For validation, the parameter settings vary slightly across methods. We adhere to the configurations in (Sun et al., 2024), as detailed in Table A1. For other methods, we primarily use the results reported in their paper. If a relevant experiment is unavailable, we generate the distilled dataset using their code and validate it under the same settings as ours.

| Parameter | CIFAR-10 | CIFAR-100 | Tiny-ImageNet | ImageNet-1K |
|---|---|---|---|---|
| Optimizer | | | AdamW | |
| Learning Rate | | | 0.01 | |
| Weight Decay | | | 0.01 | |
| Batch Size | | | 128 | |
| Augmentation | | RandomResizedCrop + Horizontal Flip | | |
| LR Scheduler | | | CosineAnneal | |
| Tempreture | | | 20 | |
| Epochs | 400 | 400 | 300 | 300 |

*Table A1.* **Hyper-parameter setttings used for our validation.**

# C. Additional Experiments

## C.1. Validate the Limitations of D$^4$M

We use D$^4$M as an example here to validate the validity of our analysis in Section 3.3 by visualizing the latents in the VAE space. As illustrated in Figure A1, we visualize the cluster centers in the VAE encoding space, which are treated as representative latents of the method, and marked as ✘. It is evident that there are many outliers in the VAE space, and the K-means algorithm treats these outliers as a separate cluster. Consequently, the latents generated by D$^4$M become concentrated around these outliers, leading to an **inaccurate representation of the VAE distribution**. Besides, each individual ✘ aims to map a single cluster, **rather than considering the entire subset of representative latents as a whole.** Moreover, ★ demonstrates that after adding noise and removing noise, **the latents undergo a shift from ✘**. Therefore, these results align well with our previous discussion.

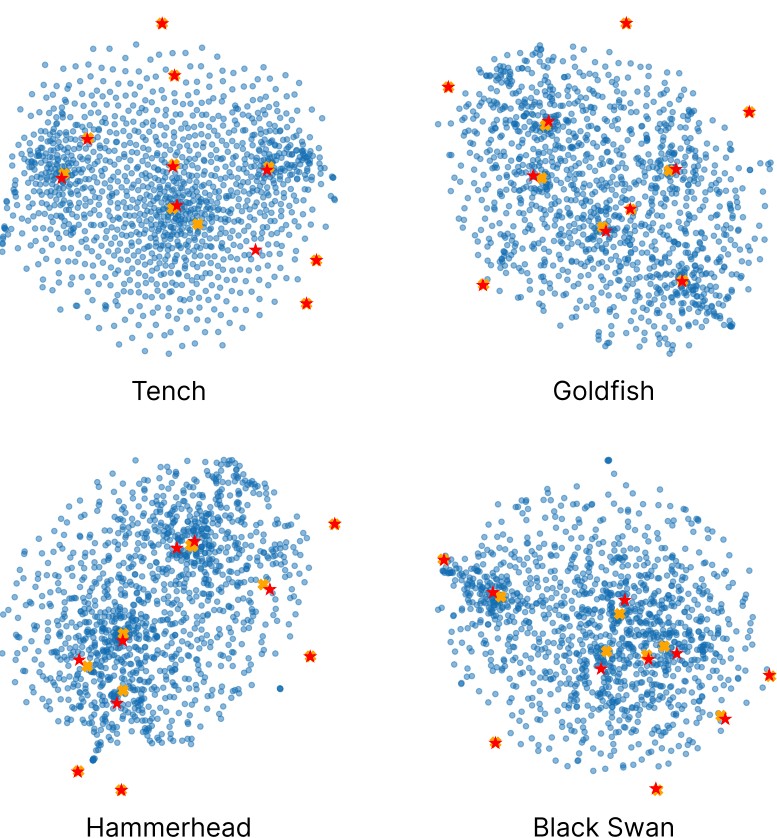

Tench                                          Goldfish

Hammerhead                                     Black Swan

*Figure A1.* **Illustration of representation-shift in D$^4$M for different classes, ImageNet-1K,** $IPC = 10$**.** The blue dots indicate the latents in the VAE space. ✘ represent the cluster centers after the VAE encoding. ★ denote the latents after adding noises and removing noises for decoding the distilled images.

## C.2. More Analysis about Group Sampling

**Computational efficiency**. We report the runtime of the group sampling process across different $m$ and different GPUs in the table, demonstrating its simplicity and efficiency. As discussed in Lines 302-306, left column, this is because multiple subsets can be sampled in parallel on the GPU instead of sequential sampling, and the operations such as random sampling and mean/variance computations are very lightweight and efficient in current computation frameworks, making the entire process highly efficient.

| IPC | 1 | 5 | 10 | 20 | 50 |
|---|---|---|---|---|---|
| Time (s) | 0.3901±0.0057 | 1.0288±0.0092 | 1.8352±0.0012 | 3.506±0.0046 | 8.5252±0.0055 |

*Table A2.* **Runtime on A100 40G with** $m = 1e6$**.**

| IPC | 1 | 5 | 10 | 20 | 50 |
|---|---|---|---|---|---|
| Time (s) | 0.1135±0.0016 | 0.1442±0.0013 | 0.2201±0.0003 | 0.3806±0.0046 | 0.9013±0.0060 |

*Table A3.* **Runtime on A100 40G with** $m = 1e5$**.**

| IPC | 1 | 5 | 10 | 20 | 50 |
|---|---|---|---|---|---|
| Time (s) | 0.0902±0.0088 | 0.1670±0.0142 | 0.2783±0.0018 | 0.5227±0.0073 | 1.2873±0.0023 |

*Table A4.* **Runtime on A6000 with** $m = 1e5$**.**

**Increasing/decreasing Variance**. As discussed in Section 4.4, we generate synthetic samples following the real data distribution, to ensure the training performance on synthetic samples. However, fow low IPCs, it is not feasible to fully match the original distribution. An intuitive approach is to increasing/decreasing the variance yields a better target distribution. We perform the experiment to verify the influence of variance. As shown in Table A5, we adjust the variance of the distribution by $50\%$ in sampling. The results show that neither increasing nor decreasing the variance leads to higher accuracy. Therefore, changing the variance may lead to a distribution deviation with samples not similar to real ones and degraded training performance.

| $-50\%$ | $-30\%$ | $-10\%$ | 0 | $+10\%$ | $+30\%$ | $+50\%$ |
|---|---|---|---|---|---|---|
| 39.8±0.5 | 42.2±0.4 | 43.1±0.4 | 44.1±0.3 | 41.4±0.4 | 37.4±0.5 | 34.0±0.6 |

*Table A5.* **Impact of increasing/decreasing variance in group sampling.**

### C.3. Visualization of Storage requirements

We present the storage requirements of different settings as discussed in Section 6.7 here. It can be seen in Figure A2, the statistical parameters are quite small, requiring only about $0.016$ GB, which includes 320 MB from the VAE weights. The storage space needed for the distilled dataset scales linearly with the increase in IPCs. When the IPC is large, the storage demand becomes significantly high. However, the combined storage requirement for the DiT weights and statistical parameters remains smaller than that of a distilled dataset with 50 IPC, yet it encapsulates all the essential information from the distillation process, regardless of IPC size. Thus, this approach offers a highly efficient way to reduce storage consumption, especially for large IPCs. Moreover, it can even encapsulate more images than the original dataset for improved training performance.

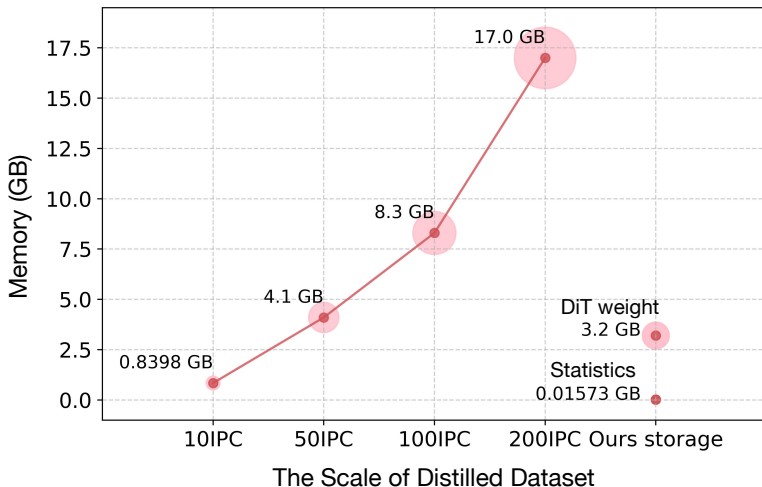

*Figure A2.* **The storage requirements of different settings on ImageNet-1K.**

## C.4. Evolution of the Latent Space Across Timesteps

We analyze the impact of different inversion timesteps on accuracy in Section 6.2. Here, we provide the visualization of the latent space transformation for different classes as inversion timestep increase Figure A3. As demonstrated in Appendix A.2, the latent space progressively transforms into a high-normality Gaussian space with the increasing timesteps.

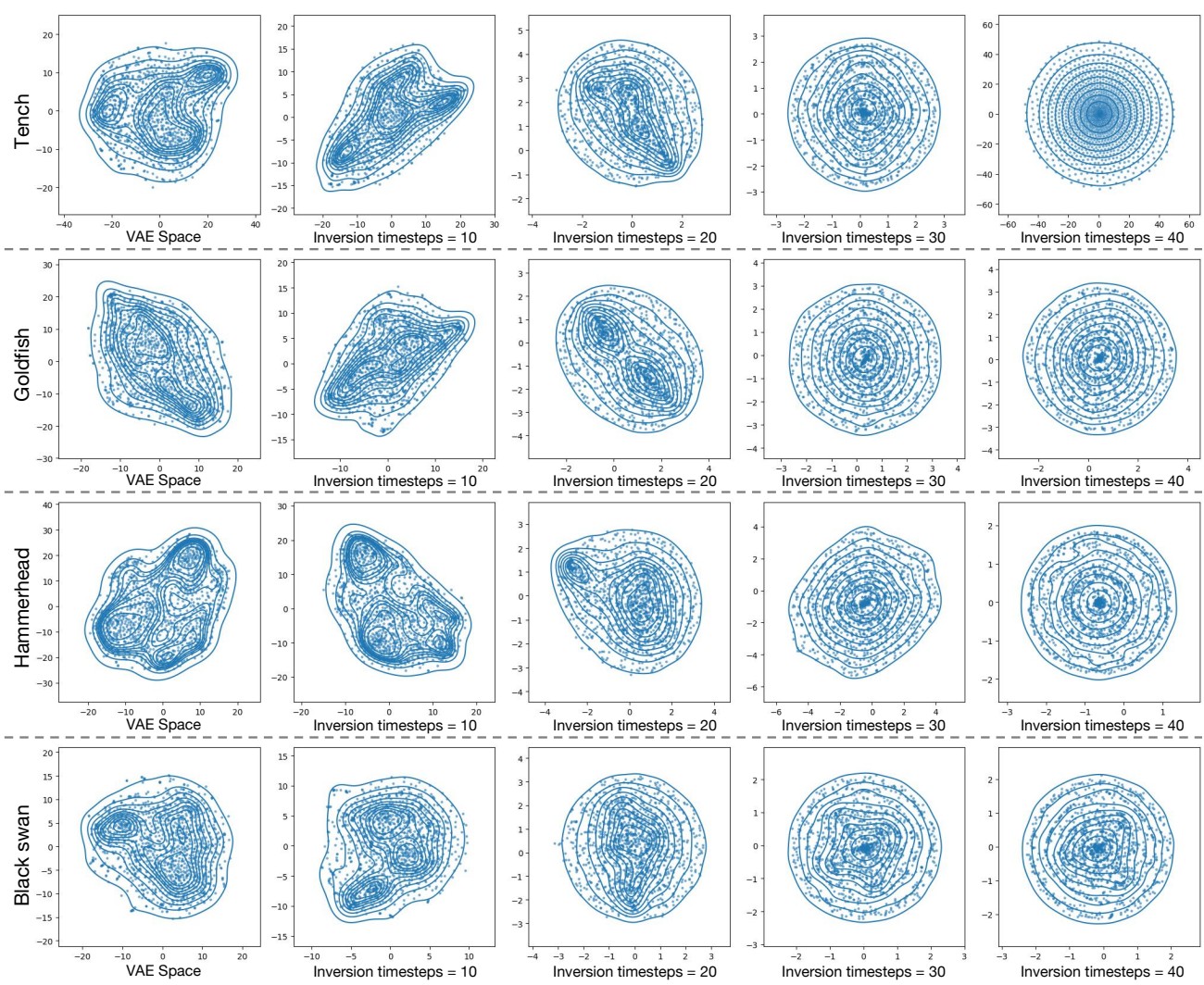

*Figure A3.* **T-SNE visualization of latent space at different inversion timesteps.**

### C.5. More Comparison of Other SOTA Methods under Different Setting

Some SOTA methods adopt different validation settings. Specifically, they train their validation models for 1000 epochs on CIFAR-10 and CIFAR-100, whereas our main comparison uses 400 training epochs. For completeness, we provide a comparison under their settings on both CIFAR and ImageNet-1K, as shown in Tables A6 and A7. Our method consistently outperforms these approaches under their setup.

| Model | Dataset | IPC | G-VBSM | ESFR | HMDC | Ours |
|-------|---------|-----|--------|------|------|------|
| ResNet-18 | CIFAR10 | 10 | 53.5±0.6 | 56.9±0.5 | 69.8±0.4 | **69.8±0.5** |
| | CIFAR10 | 50 | 59.2±0.4 | 68.3±0.3 | 75.8±0.6 | **85.2±0.4** |
| | CIFAR100 | 50 | 65.0±0.5 | - | - | **67.3±0.4** |
| ConvNetW128 | CIFAR10 | 10 | 46.5±0.7 | - | - | **55.2±0.5** |
| | CIFAR10 | 50 | 54.3±0.3 | - | - | **66.8±0.4** |
| | CIFAR100 | 50 | 45.7±0.4 | - | - | **52.1±0.5** |
| ConvNet | CIFAR10 | 10 | - | **59.9±0.2** | 47.5±0.7 | 57.3±0.4 |
| | CIFAR10 | 50 | - | 69.0±0.2 | 52.4±0.1 | **69.3±0.4** |
| | CIFAR100 | 50 | - | 51.3±0.4 | - | **54.6±0.4** |

*Table A6.* **Comparison with more SOTA methods on CIFAR-10 and CIFAR-100 (1000-epoch validation).**

| Model | IPC | G-VBSM | Teddy(post) | Teddy(prior) | Ours |
|-------|-----|--------|-------------|--------------|------|
| ResNet-18 | 10 | 31.4±0.5 | 32.7±0.2 | 34.1±0.1 | **44.3±0.3** |
| | 50 | 51.8±0.4 | 52.5±0.1 | 52.5±0.1 | **59.4±0.1** |
| | 100 | 55.7±0.4 | 56.2±0.2 | 56.5±0.1 | **62.5±0.0** |
| ResNet-101 | 10 | 38.2±0.4 | 40.0±0.1 | 40.3±0.1 | **52.1±0.4** |
| | 50 | 61.0±0.4 | – | – | **66.1±0.1** |
| | 100 | 63.7±0.2 | – | – | **68.1±0.0** |

*Table A7.* **Comparison with more SOTA methods on ImageNet-1k.**

## C.6. Cross-architecture Generalization

We broaden our experiments in Section 5.3 by integrating diverse neural network architectures, including EfficientNet-B0 (Tan & Le, 2019), ShuffleNet-V2 (Ma et al., 2018), DeiT-Tiny (Touvron et al., 2021). It can be seen in Table A8, D$^3$HR exhibits exceptional performance across substantial different architectures, incurring only a one-time generation cost.

| Student\Teacher | | ResNet-18 | EfficientNet-B0 | MobileNet-V2 | ShuffleNet-V2 | VGG-11 | DeiT-tiny |
|---|---|---|---|---|---|---|---|
| ResNet-18 | RDED | $42.3 \pm 0.6$ | $\mathbf{31.0 \pm 0.1}$ | $40.4 \pm 0.1$ | $43.3 \pm 0.3$ | $36.6 \pm 0.1$ | $25.7 \pm 0.7$ |
| | Ours | $\mathbf{44.2 \pm 0.3}$ | $30.6 \pm 0.2$ | $\mathbf{42.3 \pm 0.7}$ | $\mathbf{44.7 \pm 0.1}$ | $\mathbf{38.3 \pm 0.2}$ | $\mathbf{28.0 \pm 0.4}$ |
| EfficientNet-B0 | RDED | $42.8 \pm 0.5$ | $33.3 \pm 0.9$ | $43.6 \pm 0.2$ | $50.5 \pm 0.6$ | $35.8 \pm 0.5$ | $32.9 \pm 0.4$ |
| | Ours | $\mathbf{50.3 \pm 0.2}$ | $\mathbf{42.0 \pm 0.6}$ | $\mathbf{54.0 \pm 0.1}$ | $\mathbf{56.3 \pm 0.1}$ | $\mathbf{43.3 \pm 0.1}$ | $\mathbf{36.8 \pm 0.4}$ |
| MobileNet-V2 | RDED | $34.4 \pm 0.2$ | $24.1 \pm 0.8$ | $33.8 \pm 0.6$ | $48.2 \pm 0.5$ | $28.7 \pm 0.2$ | $24.9 \pm 0.6$ |
| | Ours | $\mathbf{43.4 \pm 0.3}$ | $\mathbf{31.8 \pm 0.8}$ | $\mathbf{46.4 \pm 0.2}$ | $\mathbf{50.0 \pm 0.2}$ | $\mathbf{37.8 \pm 0.4}$ | $\mathbf{26.5 \pm 0.8}$ |
| ShuffleNet-V2 | RDED | $37.0 \pm 0.1$ | $23.7 \pm 1.0$ | $35.6 \pm 0.2$ | $40.5 \pm 0.0$ | $29.4 \pm 0.3$ | $21.8 \pm 0.6$ |
| | Ours | $\mathbf{37.2 \pm 0.0}$ | $\mathbf{25.8 \pm 0.3}$ | $\mathbf{38.5 \pm 0.4}$ | $\mathbf{44.1 \pm 0.3}$ | $\mathbf{32.6 \pm 0.6}$ | $\mathbf{23.1 \pm 0.3}$ |
| VGG-11 | RDED | $22.7 \pm 0.1$ | $16.5 \pm 0.8$ | $21.6 \pm 0.2$ | $25.7 \pm 0.4$ | $23.5 \pm 0.3$ | $\mathbf{17.6 \pm 0.4}$ |
| | Ours | $\mathbf{25.7 \pm 0.4}$ | $\mathbf{20.2 \pm 0.2}$ | $\mathbf{24.8 \pm 0.4}$ | $\mathbf{29.1 \pm 0.4}$ | $\mathbf{28.1 \pm 0.1}$ | $15.1 \pm 0.2$ |
| DeiT-tiny | RDED | $13.2 \pm 0.3$ | $12.6 \pm 0.5$ | $13.6 \pm 0.4$ | $16.1 \pm 0.3$ | $11.4 \pm 0.1$ | $\mathbf{15.4 \pm 0.3}$ |
| | Ours | $\mathbf{17.1 \pm 0.3}$ | $\mathbf{14.8 \pm 0.2}$ | $\mathbf{17.9 \pm 0.7}$ | $\mathbf{21.4 \pm 0.8}$ | $\mathbf{14.6 \pm 0.5}$ | $15.0 \pm 0.3$ |

*Table A8.* **Comparison of Top-1 accuracy for cross-architecture generalization on ImageNet-1K,** $IPC = 10$**.**

## C.7. More Ablation Studies on the Hyper-parameters

**Choice of** $m$ **across Different Datasets.** As shown in Tables A9 and A10, we present the performance of varying $m$ on datasets of varying scales. As expected, increasing $m$ improves accuracy, as a larger candidate pool offers greater diversity and a higher chance of including representative sets. However, when $m$ becomes sufficiently large, the performance gains plateaus—indicating that the marginal benefit of adding more candidates diminishes, as the top-performing candidates become increasingly similar. We choose $m$ at the saturation point, typically between $1e5$ and $1e7$, which can work well across different datasets.

| $m$ | 1 | 1e3 | 1e4 | 1e5 | 5e5 | 1e6 | 5e6 |
|---|---|---|---|---|---|---|---|
| Accuracy (%) | $40.2 \pm 0.4$ | $40.9 \pm 0.4$ | $41.7 \pm 0.5$ | $42.4 \pm 0.4$ | $43.8 \pm 0.3$ | $44.2 \pm 0.2$ | $44.1 \pm 0.3$ |

*Table A9.* **The results across different** $m$ **under** $IPC = 10$ **on Tiny-ImageNet.**

| $m$ | 1 | 1e3 | 1e5 | 1e6 | 5e6 | 1e7 | 5e7 |
|---|---|---|---|---|---|---|---|
| Acc (%) | $38.9 \pm 0.3$ | $39.6 \pm 0.2$ | $40.4 \pm 0.3$ | $41.0 \pm 0.1$ | $41.8 \pm 0.1$ | $41.9 \pm 0.2$ | $42.1 \pm 0.2$ |

*Table A10.* **The results across different** $m$ **under** $IPC = 10$ **on CIFAR-10.**

**Different Combination of** $\lambda_\mu, \lambda_\sigma, \lambda_{\gamma_1}$**.** For $\lambda_\mu, \lambda_\delta$ and $\lambda_\gamma$, we set them to make the corresponding metrics on the same scale. We provide results for different $\lambda$ in Appendix C.7, which shows that the performance on $L_{T,C}$ is relatively robust to $\lambda$.

| Setting | 1:1:0.5 | 1:1:1 | 1:1:2 | 1:0.5:0.5 |
|---|---|---|---|---|
| Acc (%) | $44.2 \pm 0.1$ | $44.0 \pm 0.4$ | $43.6 \pm 0.3$ | $43.4 \pm 0.2$ |

*Table A11.* **The results of different combination of** $\lambda_\mu, \lambda_\sigma, \lambda_{\gamma_1}$**.**

## D. Image Visualization

We present more visualization results of the distilled images in this section. As shown in Figure A4 and Figure A5, D$^3$HR generates diversity, high-quality images for each class, effectively representing the full dataset.

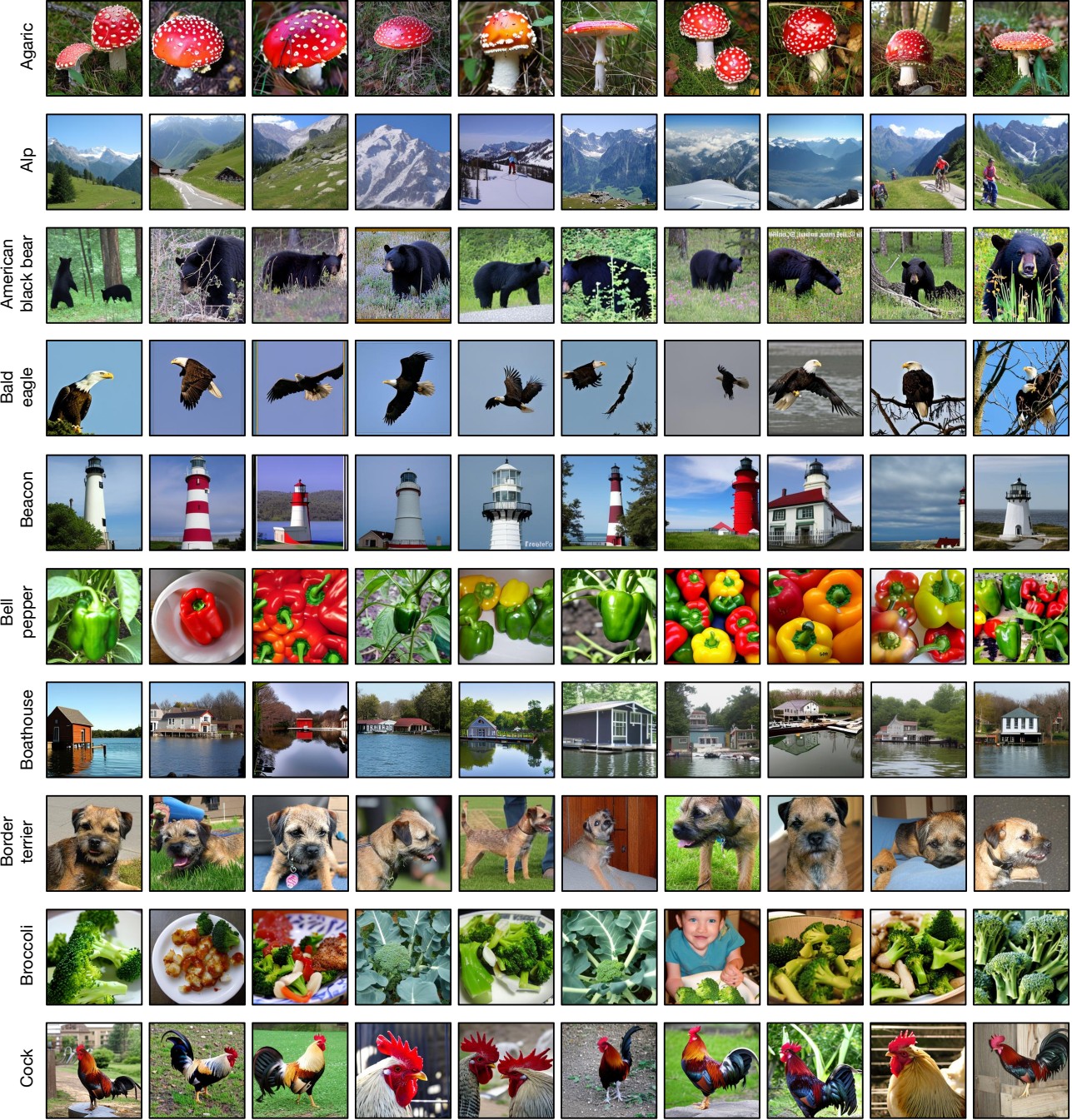

*Figure A4.* **Visualization of the distilled images for different classes on ImageNet-1K,** $IPC = 10$**, and the resolution is** $224 \times 224$**.**

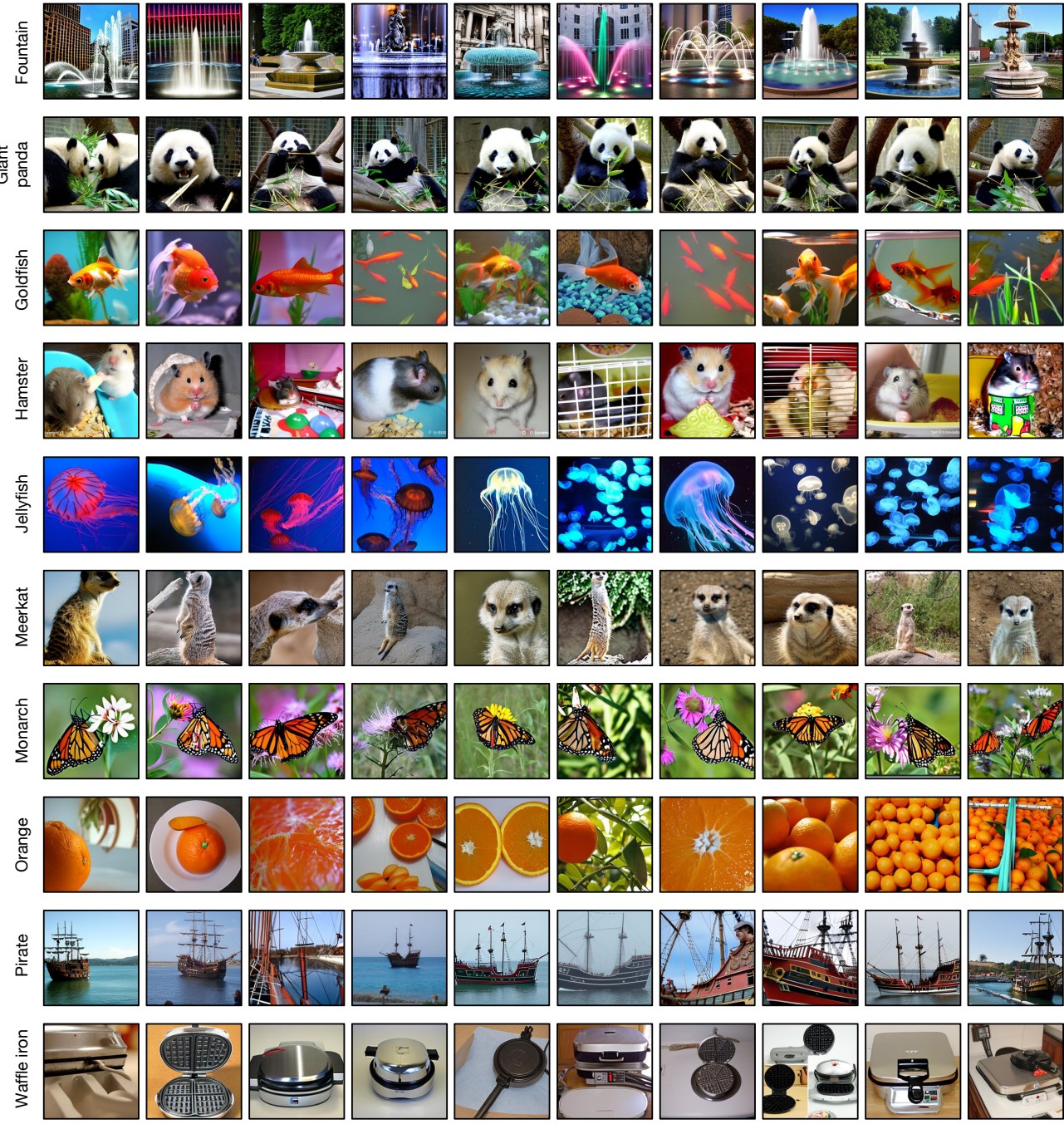

*Figure A5.* **Visualization of the distilled images for different classes on ImageNet-1K,** $IPC = 10$**, and the resolution is** $224 \times 224$**.**

