# OpenReview forum: "Taming Diffusion for Dataset Distillation with High Representativeness"
_ICML.cc/2025/Conference — ICML 2025 poster_

### Official Review · Reviewer_YwBU · 2025-03-09

**Overall Recommendation:** 4

**Summary:**

This paper utilizes DDIM inversion to map the VAE latent space into a high-normality Gaussian space. Numerous subsets are then sampled from this Gaussian distribution. The final selection is determined by identifying the subset whose distribution has the smallest loss compared to the Gaussian distribution.
The results improve across different datasets.

**## update after rebuttal:**
The author has addressed my concerns regarding the theoretical aspects and the results on CIFAR-10/100 are also greater than the state-of-the-art. I will increase my current score to Accept.

**Claims And Evidence:**

The claims made in the submission supported by clear and convincing evidence.

**Essential References Not Discussed:**

The reference of dataset distillation and diffusion models are discussed.
The authors incorporate a pre-trained Diffusion Transformer (DiT) in their implementation details section line 302, but do not specify which VAE model is used. The described process appears to involve encoding input data, mapping the latent space, sampling new latent representations, and decoding back to the original data domain. However, the role of diffusion models in this framework remains unclear.

**Experimental Designs Or Analyses:**

The current experimental designs or analyses are soundness and validity, but there are several issues can be discussed.
1. In table 1, the results on CIFAR-10 do not appear to be state-of-the-art, as multiple existing works have reported higher performance.
2. The author does not provide results for IPC = 1 on CIFAR-10 and CIFAR-100 to compare with state-of-the-art methods.

**Methods And Evaluation Criteria:**

Proposed methods make sense. The domain mapping technique efficiently generates synthetic samples that accurately capture the underlying data distribution.
All benchmark datasets are used in the dataset distillation problem.

**Other Comments Or Suggestions:**

I have no other comments.

**Other Strengths And Weaknesses:**

There are several strengths:
1. The approach demonstrates improved performance, particularly on large-scale datasets, by leveraging a structured latent space for effective sample generation
2. The method only needs a one-time generation cost, making it computationally efficient compared to iterative optimization-based distillation techniques.

**Questions For Authors:**

Why the author did not compare the results with other state-of-the-art methods.
The reason for ImageNet-1K can be they handle subsets, but most other approaches can be compared for small-scale datasets CIFAR10 and CIFAR100.

**Relation To Broader Scientific Literature:**

The use of DDIM inversion in this work is closely related to recent advancements in generative modeling that focus on improving sampling efficiency and controllability within the latent space.

**Theoretical Claims:**

About the Lemma A.1 in line 577-578, the author claim: for each class C containing m images, Zc is modeled as a discrete distribution, where the m latents are independently sampled from m distinct Gaussian distributions.
this statement is not necessarily true because it assumes independence between latent variables for each image within a class. However, in most cases, VAEs do not enforce such a strict assumption. The reason is latents are not independently sampled per image and their distributions are not necessarily distinct for each image. If the model captures class-level semantics well, the means of similar images may cluster together in the latent space. Also, In a standard VAE, the encoder learns a continuous latent space rather than modeling a discrete distribution for each class.
Thus, the claim about a multi-component Gaussian mixture distribution might be incorrect.

---

> ### Author Rebuttal · Authors · 2025-03-31
>
> **Q1: The concern about the multi-component Gaussian mixture distribution assumption.**
>
> A1: Thank you for highlighting this. To clarify the assumption, we offer a more comprehensive explanation here.
> For current diffusion-based methods and ours, the pre-trained VAE are not conditional VAE, so both the training and inference processes of the VAE are not conditioned on the class label C. In theory, each latent variable is encoded by independent statistical parameters (mean and variance), resulting in an N-component Gaussian mixture distribution for N images. In practice, as shown in Equation (10) (in Appendix), the VAE loss includes a KL divergence term, which pulls each latent distribution moderately toward N(0, I). This results in a Gaussian mixture with overlapping components, indicating that the overall mixture distribution can be well-approximated by M (< N) effective components.
>
> We give the visualization of the VAE space in the first column in Appendix B.4-Figure A4. Although the number of components is smaller than N, it remains challenging to fit the distribution, which is not in conflict with our  method. We will add the above analysis in the updated version.
>
> **Q2: More comparison with other SOTA methods on CIFAR-10, CIFAR-100.**
>
> A2: We did not include comparisons with more SOTA results on CIFAR because the methods adopt varying validation settings, making us unable to perform direct comparisons. **Since our method is geared towards more practical scenarios involving large-scale datasets**, we follow the validation setting commonly adopted by these methods that support large datasets—training for 400 epochs on CIFAR.
>
> For completeness, we also provide a comparison with [a], which achieves SOTA results on CIFAR and is trained for 1000 epochs in their paper. We directly obtained the 10 and 50 IPC distilled subsets from their official GitHub repository and evaluated them using the same validation code as in our setting to ensure a fair comparison. Our results outperform theirs under the same setup. We will include this comparison in the updated version.
>
> | Dataset  | IPC |\[a\] (ResNet-18)  | Ours (ResNet-18) | \[a\] (ResNet-101) | Ours (ResNet-101) |
> |-|-|-|-|-|-|
> | CIFAR10  | 10  | 36.4±0.3 | 41.3±0.1  | 30.4±0.6 | 35.8±0.6   |
> | CIFAR10  | 50  | 55.4±0.4 | 70.8±0.5  | 41.9±0.6 | 63.9±0.4   |
> | CIFAR100 | 10  | 46.6±0.1 | 49.4±0.2  | 31.9±0.5 | 46.0±0.5   |
> | CIFAR100 | 50  | 61.0±0.2 | 65.7±0.3  | 57.5±0.5 | 66.6±0.2   |
>
> **Q3: The experiment for IPC=1 on CIFAR-10 and CIFAR-100.**
>
> A3: We provide a comparison with RDED in the table below, which achieves excellent results on CIFAR under the 1 IPC setting.
>
> | Method| RDED (ResNet-18)| Ours (ResNet-18)| RDED (ResNet-101) | Ours (ResNet-101) |
> |-|-|-|-|-|
> | CIFAR-10  |22.9±0.4 | 24.2±0.3 | 18.7±0.1 | 21.6±0.2|
> | CIFAR-100|11.0±0.3 | 11.8±0.1 | 10.8±0.1 | 10.4±0.2|
>
> **Q4: The specific VAE model, and the role of diffusion models.**
>
> A4: Thanks for pointing this out, We adopt the pre-trained DiT and VAE from [b].
> The role of the VAE encoder in the framework is to encode the image to the latent space, while the DiT model is to inverse the latent for sampling new latent representation and then denoise the new latent representation to the VAE space. The VAE decoder is used to decode the sampled latents to pixels. Besides, as shown in Lines 149-155 left column, the motivation for using diffusion models lies in their ability to enhance realism and cross-model generalization, which are the key qualities of a high-quality distilled dataset. Based on this, as described in Lines 110–164 right column, our main motivation is that we identify three key limitations in current diffusion-based methods, primarily because of conducting optimization within the VAE latent space. This motivates us to explore a more effective optimization space, aiming to provide a better paradigm for diffusion-based methods.
> We will include further clarification in the updated version.
>
> [a] Shao, Shitong, et al. "Generalized large-scale data condensation via various backbone and statistical matching." CVPR. 2024.
>
> [b] Peebles, William, and Saining Xie. "Scalable diffusion models with transformers." ICCV. 2023.

---

> > ### Comment · Reviewer_YwBU · 2025-04-07
> >
> > The author has addressed my concerns regarding the theoretical aspects. However, the results on CIFAR-10/100 are significantly lower than the state-of-the-art. Given that this method is better suited for large-scale datasets like ImageNet-1K, I will maintain my current score.

---

> > > ### Author Response · Authors · 2025-04-09
> > >
> > > Thanks for your feedback! We would like to emphasize that **our method achieves state-of-the-art performance on the CIFAR datasets**. (i) As we mentioned in Q2, the evaluation settings on CIFAR datasets of different SOTA methods are different. Thus, it is unfair to copy their results from their papers and make direct comparisons, as the results are obtained in different settings. To make a fair comparison, we adopt a more common evaluation setting (such as train 400 epochs) from large-scale datasets to evaluate the performance on CIFAR datasets for our method and baselines,  to ensure fairness and consistency in comparison. Due to the time limitation, we finished the experiments of some SOTA baselines and show the fair comparison under the same setting in Table 1 of the paper. Our method achieves non-marginal improvements on CIFAR than SOTA baselines including D$^4$M and RDED.
> > >
> > > (ii)  Following your suggestion, we further compare our method with the SOTA method G-VBSM[a] using our evaluation setting (including 400 epochs training) as discussed above in (i). The results are shown in the response to Q2, demonstrating our superior performance than G-VBSM[a].
> > >
> > > (iii) To demonstrate the general performance of our method with a more comprehensive comparison, we further present the results of different methods using the evaluation setting and code  (including 1000 epochs training) from G-VBSM[a] to **ensure a strictly fair comparison under all the same settings**. The missing values marked with ‘-’ are unavailable because the results are not reported in the paper. The following table shows that our method can still outperform all SOTA baselines in a different evaluation setting on CIFAR for all methods with fair comparisons.
> > >
> > > R18: ResNet18; CW128: ConvNetW128; Conv: ConvNet
> > > |Dataset|IPC|G-VBSM\[a\](R18)|ESFR\[b\](R18)|HMDC\[c\](R18)|Ours(R18)|G-VBSM\[a\](CW128)|ESFR\[b\](CW128)|HMDC\[c\](CW128)|Ours(CW128)|G-VBSM\[a\](Conv)|ESFR\[b\](Conv)|HMDC\[c\](Conv)|Ours(Conv)|
> > > |-|-|-|-|-|-|-|-|-|-|-|-|-|-|
> > > | CIFAR10 |10| 53.5±0.6 | 56.9±0.5 |69.8±0.4 |**69.8±0.5** |  46.5±0.7 | - | - |**55.2±0.5** |- |**59.9±0.2** |47.54±0.7 |57.3±0.4|
> > > | CIFAR10 |50| 59.2±0.4 | 68.3±0.3 |75.8±0.6  |**85.2±0.4** | 54.3±0.3 | -| -|**66.8±0.4** | - | 69.0±0.2 |52.4±0.1 |**69.3±0.4** |
> > > | CIFAR100 | 50 | 65.0±0.5 |-|-|**67.3±0.4** |45.7±0.4 | -|-|**52.1 ±0.5** |- |51.3±0.4 |-| **54.6±0.4** |
> > >
> > > To summarize, our method achieves SOTA performance on CIFAR. In our evaluation setting, our method performs the best compared with SOTA baselines including RDED (shown in our paper) and G-VBSM[a] (shown in the rebuttal). In another evaluation setting following  G-VBSM[a] with their code, our method can still perform the best compared with SOTA baselines including [b] and [c]. We will make this more clear in our paper revision.
> > >
> > >
> > > [a] Shao, Shitong, et al. "Generalized large-scale data condensation via various backbone and statistical matching." CVPR. 2024.
> > >
> > > [b] Deng, Wenxiao, et al. "Exploiting inter-sample and inter-feature relations in dataset distillation." CVPR. 2024.
> > >
> > > [c] Moon J Y, Kim J U, Park G M. “Towards Model-Agnostic Dataset Condensation by Heterogeneous Models” ECCV. 2024.

---

### Official Review · Reviewer_phyT · 2025-03-11

**Overall Recommendation:** 4

**Summary:**

This paper addresses limitations in diffusion-based dataset distillation methods and introduces D3HR, a novel framework that enhances the representativeness of distilled datasets. This paper reveal that current methods suffer from issues like inaccurate distribution matching, distribution deviation due to random noise, and separate sampling, leading to suboptimal performance. The proposed D3HR framework outperforms existing state-of-the-art dataset distillation methods across multiple datasets, including CIFAR-10, CIFAR-100, Tiny-ImageNet, and ImageNet-1K. It demonstrates better generalization across architectures like ResNet, MobileNet, and VGG.

## update after rebuttal
The authors' response has addressed my concerns, so I maintain my overall positive assessment.

**Claims And Evidence:**

The claims are supported by clear and convincing evidence.

**Essential References Not Discussed:**

No.

**Experimental Designs Or Analyses:**

The experimental designs are reasonable.

**Methods And Evaluation Criteria:**

Yes.

**Other Comments Or Suggestions:**

In Equation (2), the variable C is not explicitly defined, which may cause confusion for the reader. Clarifying its meaning within the equation or referring to its definition elsewhere in the text would improve clarity.

**Other Strengths And Weaknesses:**

Strengths:

-	The proposed method improves distribution matching. Uses DDIM inversion to transform latents into a more Gaussian-like distribution, enhancing representativeness.

-	The proposed method achieves state-of-the-art performance on different neural architectures.

-	This work propose an efficient sampling strategy to reduce randomness in dataset generation and ensures that distilled samples align better with the original distribution.

Weaknesses:

-	Figures 1 and 3 appear to be less informative. Even after carefully reading the paper, their meaning remains unclear. Additionally, in the caption of Figure 1, the "blue lines" mentioned cannot be found, which may cause confusion. Providing more detailed explanations or clearer visual indicators would improve their interpretability.

-	This work highlights the issue of inaccurate distribution matching in the latent space found in previous methods. However, it is not clearly explained why the proposed method achieves better distribution alignment. A more in-depth discussion or empirical validation of how the mapping improves distribution matching would strengthen the argument.

-	The paper lacks an ablation study on domain mapping and group sampling. Since these are key components of the proposed method, conducting and presenting ablation experiments would help validate their individual contributions to performance improvements.

-	SRe2L relies on a large amount of soft labels to enhance dataset distillation performance. It would be valuable to evaluate the effectiveness of the proposed method using hard labels. This would provide a better measure of the quality of the generated images, independent of external supervisory signals.

**Questions For Authors:**

In Figure 5, the performance appears to be significantly influenced by the number of inversion timesteps. It would be helpful to discuss why this dependency occurs and whether there is an optimal range of timesteps that balances performance and computational efficiency.

Regarding storage in Section 6.6, does the proposed method also require storing the decoder D? If so, how does this affect the overall storage efficiency? Clarifying whether the decoder needs to be retained separately or if it can be reconstructed from stored parameters would provide a more complete picture of the method’s storage requirements.

**Relation To Broader Scientific Literature:**

D3HR advances the field by improving distribution alignment, reducing noise-induced artifacts, and enhancing dataset compression efficiency, making it a significant step forward in scalable dataset distillation methods.

**Theoretical Claims:**

I have checked the correctness of the proofs for theoretical claims.

---

> ### Author Rebuttal · Authors · 2025-03-31
>
> **Q1: Figure 1 and 3 less informative**
>
> A1: Thanks for the suggestions. The “blue lines” are the “blue contour lines” which indicate the probability density of the distribution. We will revise the caption to clarify this.
> Figure 1 is intended to convey two key messages: (1) The VAE latent space exhibits significantly lower normality compared to our mapped (inversion) space. The “blue contour lines” allow us to clearly observe the structure and concentration of the latents (blue dots). Denser and more centralized contours reflect regions with higher latent density, which are the areas we should pay more attention to. (2) We visualize representative latent points generated by our method. In Figure 1(b), we show the latents sampled in the mapped (high-normality) space, demonstrating that our sampling process closely matches the desired noise distribution. To further validate the fidelity of the mapping, Figure 1(a) presents the corresponding latents in the original VAE space after DDIM sampling. The blue contour lines help illustrate that our sampling process successfully captures the structure of the VAE latent space, and effectively concentrates in high-density regions.
> Similarly, Figure 3 visualizes n latents in the VAE space—comparing those generated by the default DiT model and our proposed method. It demonstrates that our approach yields more representative and diverse latents.
> We will provide more detailed explanations and clearer visual indicators to improve interpretability.
>
> **Q2: Deep discussion or empirical validation on why better alignment can be achieved in the inversion space**
>
> A2: The reason behind this is that a Gaussian mixture in the VAE space is more difficult to fit than a single Gaussian distribution in the inversion space. This is because compared to a Gaussian mixture with an uncertain number of components and unknown structure, a single Gaussian is a simpler and well-defined parametric form.
>
> To prove our method achieves better distribution alignment, we visualize 10 generated latents in the VAE space for the same class ("Goldfish") using both D4M and ours, as shown in Appendix Figure A1 and Figure 1 in the main paper. The results show that D4M, which performs optimization directly in the VAE space, tends to select latents near the edges of the distribution. In contrast, our method produces a more representative and diverse set of latents, better capturing the overall structure of the distribution.
> For the quantitative comparison of accuracy, we refer to three key results: the D4M in Table 1, the default DiT generation in Table 3 (row 1), and our method with domain mapping but without group sampling in Table 3 (row 2). Together, the last has the highest accuracy, which demonstrates the effectiveness and necessity of our domain mapping strategy.
> Finally, to further support our claims, we also include a visual comparison of generated images for D4M, the default DiT, and ours. These qualitative results clearly illustrate the advantages of our approach in producing more representative samples.
>
> **Q3: Ablation study on domain mapping and group sampling**
>
> A3: We have the ablation studies in Section 6.1.
>
> For the group sampling, we present the results in Table 3 (rows 2–9). These include **our method without group sampling (row 2)**, as well as ablations that include only partial components of $L_{T, C}$. This allows us to verify the importance of group sampling and to assess the individual contributions of each component in $L_{T, C}$.
>
> For the domain mapping, as discussed in lines 366–374, left column, we use the results in Table 3 (rows 1–2) to verify its importance. We compare **the default DiT generation with DDPM (row 1)** and **domain mapping with DDIM (row 2)**, with the same configuration for all other steps. The results clearly demonstrate the advantages of domain mapping with DDIM (Line 375-384 left column).
> We will give a more clear description and annotation.
>
> **Q4: Performance on hard labels**
>
> A4: Please refer to Q1 of the response for xaqL.
>
> **Q5: Clarifying C for Equation 2**
>
> A5: C denotes the class condition, which means the same as other equations. We will provide a clearer explanation of this in the updated version.
>
> **Q6: Discussion of different Inversion Timesteps**
>
> A6: As discussed in Lines 378–415, left column, there is a trade-off between maintaining the Gaussian assumption and preserving image structural information across different time steps t. Please refer to Q3 of the response for xB8L with more details.
> The choice of inversion steps is guided by empirical observations. As shown in Figure 5, using 24–31 inversion steps consistently achieves SOTA accuracy, making it a reasonable and effective choice.
>
> **Q7: Storage of decoder**
>
> A7: In Appendix Figure A3,  the decoder size has already been accounted for in our computation under “DiT weight.” The storage size of the VAE weights is 320MB. We will include this detail more clearly.

---

### Official Review · Reviewer_xB8L · 2025-03-13

**Overall Recommendation:** 3

**Summary:**

This article introduces the D³HR framework (Taming Diffusion for Dataset Distillation with High Representativeness) to tackle the issue of inaccurate distribution matching in existing diffusion-based dataset distillation methods. D³HR enhances distribution matching accuracy by utilizing DDIM inversion to transform the VAE latent space into a high-normality Gaussian domain. Additionally, it ensures thorough alignment of dataset distributions through a group sampling strategy that incorporates statistical constraints. Extensive benchmark experiments reveal significant performance improvements compared to previous approaches.

## update after rebuttal
This paper raises the issue of inaccurate distribution matching in existing diffusion-based dataset distillation methods and proposes a solid solution to it. The response answered my concerns on experiments. Thus, I keep my original rating of weak acceptance.

**Claims And Evidence:**

Yes. The authors claim the efficient alignment of distribution of latent vectors, which is supported by results and visualization.

**Essential References Not Discussed:**

No

**Experimental Designs Or Analyses:**

Yes. The experimental design follows the common practice. More diffusion-based DD methods should be included, for example minimax diffusion.

**Methods And Evaluation Criteria:**

Yes. The proposed method is reasonable and the evaluation follows the common practice. Detailed ablation study is implemented to verify the effectiveness of the method.

**Other Comments Or Suggestions:**

Please address the weaknesses.

**Other Strengths And Weaknesses:**

Strengths:

1.The proposed method effectively resolves the issue of overall distribution deviation caused by individual sampling in previous diffusion-based dataset distillation approaches.

2.The experimental results demonstrate that the method is both effective in improving distillation quality and efficient in terms of computational performance.

Weakness:

1.The paper seems to focus more on selecting the optimal subset, i.e., choosing a subset from the original dataset that best matches the complete distribution. It seems that the method added an optimal input selection step to the diffusion-based image synthesis process.

**Questions For Authors:**

1. Does the DDIM inversion process introduce significant additional computational costs when mapping the latent space of the entire dataset to a Gaussian noise domain?

2. Section 6.2 highlights that performance degrades sharply when the DDIM inversion step count (T) is too high (e.g., T > 31). What is the root cause of this phenomenon?

3. In practice, is there potential for cross-selection between different candidate subsets in group sampling? The paper appears to select only one subset—could this lead to suboptimal global distribution alignment?

4. Does the loss function $\mathcal{L}_{T,\mathcal{C}}$ function more as a static evaluation metric rather than an optimizable loss?

5. How is the optimal number of candidate subsets (m) determined theoretically? For datasets of varying sizes, how should m be chosen?

6. The $\mathcal{L}_{T,\mathcal{C}}$ includes multiple hyperparameters (e.g., $\lambda_\mu$, $\lambda_\sigma$, $\lambda_{\gamma_1}$). Does this complexity complicate practical usage and reproducibility?

**Relation To Broader Scientific Literature:**

D^3HR considers the overall distribution when generating distillation datasets, rather than sampling them individually, thus improving the representativeness of distillation datasets.

**Theoretical Claims:**

Yes, the authors provided the theoretical analysis about the VAE latent fitting and the latent distribution in DDIM inversion in the supplementary.

---

> ### Author Rebuttal · Authors · 2025-04-01
>
> **Q1: primarily focuses on optimal subset selection**
>
> In line 027-044 right column, we identify three key limitations in diffusion-based methods due to their reliance on optimization in the VAE latent space. This motivates us to identify a more effective space by DDIM inversion, aiming to provide a better paradigm for diffusion-based methods. Our contribution lies in **the entire pipeline designed for dataset distillation, including DDIM inversion, distribution matching, group sampling, and generation**—not merely the sampling. Each component is carefully designed to cohesively operate in the pipeline to ensure that the distilled subset is both compact and highly representative.
>
> **The process in the inversion space is not input selection. It is a principled generative procedure.** As discussed in Sec. 4.3, we first map the original latent distribution to a Gaussian distribution due to Lemma 4.1, enabling efficient sampling via its high-normality. We first generate latents that **probabilistically follow the distribution of the entire class**, as shown in line 260–266 left column. Building on this, group sampling then supports parallel generation of candidate subsets (each maintaining distributional alignment), from which we identify the most representative one. The whole process tightly integrates sampling and optimization, enabling effective and distribution-aware subset generation tailored for diffusion-based synthesis, and goes beyond simple input selection.
>
> **Q2: The computation overhead for DDIM inversion**
>
> Please refer to Q6 of the response for Reviewer VDRK.
>
> **Q3: Extreme high T incur accuracy degradation**
>
> As discussed in Line 378–415 left column, there is a trade-off between maintaining the Gaussian assumption and preserving image structural information across different steps t. When t is small (such as 20), the distribution is a mixture of Gaussians as shown in Figure A4, and our distribution matching with a single Gaussian (Sec. 4.3) is not able to accurately describe the Gaussian mixtures, leading to certain performance loss. When t becomes large such as 40, although our distribution matching can accurately represent the real distributions which becomes more normal (Figure A4), the real distributions suffer from more significantly structural information loss due to adding more noise, which in turn degrades the performance of DDIM inversion. Thus, it is a trade-off between maintaining the Gaussian assumption and preserving image structural information.
>
> **Q4: Using cross-selection in group sampling**
>
> To validate cross-selection effect, we experiment on Tiny-ImageNet using 10 representative subsets with 10 IPC (performing group sampling 10 times; average accuracy: 44.07; individual results shown in Figure A2). We randomly combine the subsets, as shown in the table below, all three combined results yield lower accuracy than individual subsets. As discussed in Line 240–244 right column, this is because our $L_{T,C}$ is designed to ensure that the entire subset aligns well with the desired distribution, rather than fitting individual latents. Cross-combining latents from different subsets breaks this design principle without any distribution alignments.
>
> |Setting|Combined set1|Combined set2|Combined set3|
> |-|-|-|-|
> |Acc|42.3±0.4|42.0±0.4|41.9±0.5|
>
> **Q5: The role of $L_{T, C}$, is $L_{T,C}$ complexity for usage and reproducibilit**
>
> Indeed, $L_{T,C}$ ​ is a static evaluation metric, which is easy to compute with basic mathematical operations, and the sampling process is efficient (Please refer to Q3-(1) of the response for xaqL). For $\lambda_\mu$, $\lambda_\delta$ and $\lambda_\gamma$, we set them to make the corresponding metrics on the same scale. We provide results for different $\lambda$ in Table below, which shows that the performance on $L_{T,C}$ is relatively robust to $\lambda$.
>
> |Setting|1:1:0.5|1:1:1|1:1:2|1:0.5:0.5|
> |-|-|-|-|-|
> |Acc|44.2±0.1|44.0±0.4|43.6±0.3|43.4±0.2|
>
> **Q6: Choice of m across the datasets with different sizes**
>
> We empirically set the value of m. The table illustrates how accuracy varies with different m across datasets of different scales. As expected, increasing m improves accuracy, as a larger candidate pool offers greater diversity and a higher chance of including representative sets. However, when m becomes sufficiently large, the performance gains plateaus—indicating that the marginal benefit of adding more candidates diminishes, as the top-performing candidates become increasingly similar. We choose m at the saturation point, typically between 1e5 and 1e7, which can work well across different datasets.
>
> Tiny-Imagenet, 10IPC:
> |m|1|1e3|1e4|1e5|5e5|1e6|5e6|
> |-|-|-|-|-|-|-|-|
> |Acc|40.2±0.4|40.9±0.4|41.7±0.5|42.4±0.4|43.8±0.3|44.2±0.2|44.1±0.3|
>
> CIFAR10, 10IPC:
> |m|1|1e3|1e5|1e6|5e6|1e7|5e7|
> |-|-|-|-|-|-|-|-|
> |Acc|38.9±0.3|39.6±0.2|40.4±0.3|41.0±0.1|41.8±0.1|41.9±0.2|42.1±0.2|
>
> **Q7: Comparison with Minimax**
>
> Please refer to Q1 of the response for xaqL.

---

> > ### Comment · Reviewer_xB8L · 2025-04-07
> >
> > Thanks for the response! My main concerns about the subset selection and experimental details have been addressed. I would like to keep my original rate.

---

### Official Review · Reviewer_xaqL · 2025-03-14

**Overall Recommendation:** 3

**Summary:**

This work proposes a novel diffusion-based dataset distillation solution. Based on the fact that previous methods suffer from inaccurate distribution matching, the authors propose to convert the images to latents with DDIM inversion and model it as Gaussian. Then, sample multiple subsets from the Gaussian and select the subset with the most similar distribution statistics. These latents are sent to DDIM for image generation. The proposed method achieves leading performance on regular DD benchmarks.



**Update after rebuttal**: I appreciate the authors' feedback and some concerns are addressed. So I still leans toward positive and would keep my initial positive score.

**Claims And Evidence:**

The authors made three main claims (summarized in lines 21-44), which are reasonable and supported in the paper.

**Essential References Not Discussed:**

No missing references.

**Experimental Designs Or Analyses:**

The experiment design including the selection of teacher models, IPCs, and ablation study is sound.

**Methods And Evaluation Criteria:**

The method is constructed following the previous observations, which is sound.

The authors adopt regular benchmarks for dataset distillation, including small and large-scale image datasets, and also cross architecture protocol (appendix). The only concern is that they did not compare with MiniMax diffusion (Gu et al., CVPR 2024) which is also diffusion-based and may yield competitive performance (58.6\% on ImageNet-1K with IPC 50).

**Other Comments Or Suggestions:**

No other comments

**Other Strengths And Weaknesses:**

Other weaknesses:

1. The "group sampling" use sampling-reject paradigm to find the best latent subset, which seems be computationally heavy.

**Questions For Authors:**

I summarize my concerns as follows, including these in the previous questions for the authors convienience:

1. The experimental comparison to MiniMax (Gu et al., CVPR'24) is missing, which might be a competitive baseline.
2. For Lemma 3.1: why the Gaussian mixture is "hard-to-fit"? This is concluded from which perspective?
3. For group sampling:
    (1) It seems to be computationally heavy. Is the "2.6s per class" on 306 the time for group sampling?
    (2) Why not sample $IPC-3$ latents and "solve" the other two latent based on the three constraints to avoid repeated sampling?
    (3) This sampling method implies that the distribution of real samples is exactly the target of synthetic samples. However, this is not supported. It is possible that increasing/decreasing the variance yields a better target distribution.
    (4) For low sample capacity (IPC), the distribution statistics may be biased.

I would tune my rating if my questions are well addressed.

**Relation To Broader Scientific Literature:**

N/A

**Theoretical Claims:**

The proofs are checked with no issue found.

---

> ### Author Rebuttal · Authors · 2025-03-31
>
> **Q1: Comparison to MiniMax with hard labels**
>
> As noted in Lines 297–300 right column, we did not include Minimax in the main table as it focuses on small subsets of ImageNet-1K. For large datasets, it requires extra training of multiple diffusion models with high computational cost. They only report results for ResNet-18 on ImageNet-1K, and do not support the other three datasets and other architectures used in our table. For a fair comparison with their results, we give the comparison with Minimax using their main setting on ImageWoof. Minimax is the SOTA method with hard labels on ImageWoof. Our method with 224×224 resolution outperforms Minimax Diffusion with 256×256 resolution.
>
> |IPC|model|Minimax|Ours|
> |-|-|-|-|
> |10|resnet18|37.6±0.9|39.6±1|
> |10|resnetAP-10|39.2±1.3|40.73±1|
> |50|resnet18|57.1±0.6|57.6±0.4|
> |50|resnetAP-10|56.3±1.0|59.3±0.4|
> |100|resnet18|65.7±0.4| 66.8±0.6|
> |100|resnetAP-10|64.5±0.2|64.7±0.3|
>
> **Q2: Why is Gaussian mixture "hard-to-fit"**
>
> "hard-to-fit" means that a Gaussian mixture is much more difficult to fit than a single Gaussian, due to its uncertain number of components and unknown structure. In contrast, a single Gaussian is a simpler and well-defined parametric form. In addition, in Fig. A1, we present the visualization of fitting the Gaussian mixture using K-means clustering under the D4M setting. It can be observed that some outliers near the edges of the distribution are selected, indicating that the Gaussian mixture is not well captured by the clustering. This further supports our point that the Gaussian mixture is relatively hard to fit.
>
> **Q3: About group sampling**
>
> Thanks for the valuable feedback. We explain the group samping more clearly, and then address your comments.
> **Rationality of group sampling**: We propose an efficient sampling design based on the mapped high-normality property. As discussed in Line 266-270 left column, the distribution of n latents (for n IPC) may still deviate from the desired distribution due to limited size n. To address this, we sample multiple subsets in parallel and choose the most representative subset with the closest distribution to the desired distribution, as detailed below.
>
> **(1) Computational efficiency**: We report the runtime of the group sampling process in the table, demonstrating its simplicity and efficiency. As discussed in Lines 302-306 left column, this is because multiple subsets can be sampled in parallel on the GPU instead of sequential sampling, and the operations such as random sampling and mean/variance computations are very lightweight and efficient in current computation frameworks, making the entire process highly efficient.
>
> m = 1e5, A100 40G:
> |IPC|1|5|10|20|50|
> |-|-|-|-|-|-|
> |time(s)|0.3901±0.0057|1.0288±0.0092|1.8352±0.0012|3.506±0.0046| 8.5252±0.0055|
>
> **(2) About analytical solutions**: Group sampling is straightforward and efficient, taking just a few seconds to sample millions of subsets and select the best one. The mentioned analytical method to sample N-3 and solve the rest for accurate distribution matching may require additional complex solvers with high computation cost. Furthermore, with the solved examples, although the statistics of the distribution may match perfectly, the overall distribution of all samples may not be a Gaussian distribution. It is possible that the designed examples are far away from the sampled examples to match the desired distribution which is in the middle between them.
>
> **(3) Increasing/decreasing variance**: Our domain mapping (Sec. 4.2) and distribution matching (Sec. 4.3) try our best to build an easy-to-fit distribution of real data. Then we generate synthetic samples following the real data distribution, to ensure the training performance on synthetic samples. Changing the variance may lead to a distribution deviation with samples not similar to real ones and degraded training performance.   We further perform the experiment to verify the influence of variance. As shown in the table, we adjust the variance of the distribution by ±50% in sampling. The results show that neither increasing nor decreasing the variance leads to higher accuracy.
>
> |-50%|-30%|-10%|0|+10%|+30%|+50%|
> |-|-|-|-|-|-|-|
> |39.8±0.5|42.2±0.4|43.1±0.4|44.1±0.3|41.4±0.4|37.4±0.5|34.0±0.6|
>
> **(4) Statistics may be biased for low IPCs**: As the law of large numbers suggests, smaller sample sizes indeed leads to higher statistical bias. This also provides a reasonable explanation for the observed drop in accuracy at low IPC. This is a general issue for dataset distillation works. To mitigate this, our group sampling generates m subsets and selects the most representative one with the closest distribution. Thus, although one subset may suffer from large statistical bias, it is possible to find another subset with smaller bias. The results in Table 3 show our best performance compared with baselines in low IPC, showing the effectiveness of our method in addressing the bias issue.

---

### Official Review · Reviewer_VDRK · 2025-03-17

**Overall Recommendation:** 3

**Summary:**

This paper proposes a diffusion-based dataset distillation method. The paper claims that the VAE space of the diffusion model is more difficult for distribution matching. To tackle this challenge, the core idea of the proposed method is to apply DDIM inversion to each sample in the original dataset and then model the distribution of the inverted samples. To obtain distilled samples, the method performs latent-space sampling for multiple times and select the sample group that yields the lowest loss. Experiments on some datasets validate the effectiveness of the method.

**Claims And Evidence:**

The claims are not well supported. The authors claim that "distribution in the VAE space has low-degree of normality". However, there is no clear definition of "normality" or any reference. The visualization in Fig. 1 is not informative, either. One can hardly understand the difference between two plots. The only difference in the current version is that there are some offsets between the points.

**Essential References Not Discussed:**

Missing Reference:

[a] Generalized Large-Scale Data Condensation via Various Backbone and Statistical Matching, Shao et al., CVPR 2024.

[b] Teddy: Efficient Large-Scale Dataset Distillation via Taylor-Approximated Matching, Yu et al., ECCV 2024.

[c] Diversity-driven synthesis: Enhancing dataset distillation through directed weight adjustment, Du et al., NeurIPS 2024.

**Experimental Designs Or Analyses:**

The experiments are clear to validate the effectiveness of the proposed method over some baselines. But I am not sure if it is optimal.

**Methods And Evaluation Criteria:**

I am not sufficiently sure that the proposed method makes sense. If the target is to find a space that can effectively matching the distribution, why not consider the embedding space before the final linear layer of a pre-trained classifier, where samples in various class are almost separable. Indeed converting to the feature space of inverted samples can reduce the difficulty of distribution matching, but I am not sure whether this strategy is optimal. Although I can find some motivation about architectural scalability between Line 149 and 155, this issue can be potentially addressed by including more structures during training [a].

Besides, why use sampling-based method to select sample groups instead of conducting gradient-based optimization ?

[a] Generalized Large-Scale Data Condensation via Various Backbone and Statistical Matching, Shao et al., CVPR 2024.

**Other Comments Or Suggestions:**

There are no blue lines in Fig. 1.

**Other Strengths And Weaknesses:**

Strengths:

1. The experimental results are good and the method surpasses the diffusion-based methods.
2. The writing is generally clear.

Other Weaknesses:

1. I am afraid that the time efficiency may not be superior as claimed by the authors. First, we need to train a diffusion model on the original dataset, which can take multiple days especially on large-scale ones. Second, the DDIM inversion process for each sample is obviously non-trivial.

**Questions For Authors:**

I am also curious about the results of using hard labels, because this case can reflect the capacity of distilled samples best.

**Relation To Broader Scientific Literature:**

NA

**Theoretical Claims:**

I quickly go over the introduced lemma and the proof, and I find no issue within these parts.

---

> ### Author Rebuttal · Authors · 2025-03-31
>
> **Q1: no clear definition of normality**
>
> Thanks for pointing this out. In our context, normality refers to the degree of the latent space data conforms to a normal distribution. A higher level of normality indicates that the latent distribution more closely resembles a normal distribution. This usage is consistent with that in statistical normality test, which assess whether sample data is drawn from a normal distribution. We will add the clarification in the updated version.
>
> **Q2: not informative Figure 1**
>
> Please refer to Q1 of the response for phyT.
>
> **Q3: Why not use the embedding space of a pre-trained model**
>
> First, the scalability to multiple architectures in Lines 149-155 left column are the motivation why we use diffusion models. We agree with prior works (RDED, D4M, Minimax) that realism and cross-model generalization are key qualities of a high-quality distilled dataset, which can be effectively achieved with the help of diffusion models.
>
> Based on this, as shown in Lines 110–164 right column, our main motivation is that we identify three key limitations in current diffusion-based methods, primarily because of conducting optimization within the VAE latent space. This motivates us to explore a more effective optimization space, aiming to provide a better paradigm for diffusion-based methods.
>
> Certain previous approaches that rely on the embedding space of a pretrained model or a small set of pretrained models, can introduce architecture bias, which may limit their generalization capabilities in large scale datasets or different independent architectures. In contrast, we identify a better optimization space that is agnostic to any specific architecture, allowing a single distilled dataset to be directly used across a wide range of models for users. As shown in the Tabel A1 in the appendix, our method can achieve SOTA performance across various models just with one-time generation cost.
>
> **Q4: Sampling-based method instead of gradient-based optimization for group sampling**
>
> We propose an efficient and effective sampling design based on the mapped high-normality property. As discussed in Lines 260-266 left column, the n latents (corresponding to n IPC) probabilistically follows the distribution of the whole class C. Building on this, our group sampling method is simple and efficient to search the most representative subset. For computation overhead analysis, please refer to Q3-(1) of the response for xaqL. In contrast, the gradient-based optimization is computationally expensive and may compromise the probabilistic nature of the n latents.
>
> **Q5: The missing references**
>
> Regarding [c], we have already included the comparison in our paper under the name DWA in Table 1.
> As for [a] and [b], we provide the comparison on ImageNet-1K below, and will include the corresponding reference in the next version.
>
> Resnet 18:
> |IPC|G-VBSM|Teddy(post)|Teddy(prior)|Ours|
> |-|-|-|-|-|
> |10|31.4±0.5|32.7±0.2|34.1±0.1|44.3±0.3|
> |50|51.8±0.4|52.5±0.1|52.5±0.1|59.4±0.1|
> |100|55.7±0.4|56.2±0.2|56.5±0.1|62.5±0.0|
>
> Cross Resnet101:
> |IPC|G-VBSM|Teddy(post)|Teddy(prior)|Ours|
> |-|-|-|-|-|
> |10| 38.2±0.4|40.0±0.1|40.3±0.1| 52.1±0.4|
> |50|61.0±0.4|-|-|66.1±0.1|
> |100|63.7±0.2|-|-|68.1±0.0|
>
> **Q6: Time cost of a pre-trained diffusion model and DDIM inversion**
>
> It is common to use pre-trained models in dataset distillation. Diffusion-based methods (D4M, Minimax) rely on a pre-trained diffusion model, while other methods often use pre-trained teacher models to guide the distillation process. Thus, we do not count the training of pre-trained models towards our time cost.
>
> The DDIM inversion is effective without high computation cost. (i) As discussed in Sec. 5.3, we only need to perform DDIM inversion once with a one-time cost, to generate multiple distilled datasets for different model architectures under various IPC settings. This is different from other methods which need to run their whole algorithms one more time if the architecture or IPC setting changes. (ii) The results of DDIM inversion are easy to store with little storage requirement, as shown in Appendix-Figure A3. (iii) Moreover, the computation cost of DDIM inversion is affordable. Even for ImageNet-1K, our method only requires approximately 4.5 hours on a single node with 8 A100 GPUs. In comparison to the SOTA diffusion-based method Minimax, which requires fine-tuning 50 separate pre-trained DiTs for ImageNet-1K, our approach is significantly more efficient. In addition, if more efficient denoisers (e.g., DPM-Solver families) are used, our paradigm can be further accelerated. This may be an orthogonal improvement direction and not the focus of our work.
>
> **Q7: Comparison with hard labels**
>
> We give the comparison results with Minimax diffusion, which is the SOTA method with hard labels under their main evaluation setting on ImageWoof. Our method with 224×224 resolution outperforms Minimax Diffusion with 256×256 resolution. Please refer to Q1 of the response for xaqL.

---

> > ### Comment · Reviewer_VDRK · 2025-04-05
> >
> > Thanks for the response. My concerns are partially addressed. Indeed the experiments show improvement over existing methods. On this basis, I am happy to increase my score. But I am still concerning the efficiency, the optimality, and the motivation of the sampling-based method, which can potentially be valuable for future research.

---

> > > ### Author Response · Authors · 2025-04-09
> > >
> > > Thanks for your comment about the sampling-based method. To better address your concern, we revise the explanation for clarity.
> > >
> > > For **efficiency**, we report the runtime of the group sampling process in the table, demonstrating its simplicity and efficiency. As discussed in Lines 302-306 left column, this is because multiple subsets can be sampled in parallel on the GPU instead of sequential sampling, and the operations such as random sampling and mean/variance computations are very lightweight and efficient in current computation frameworks, making the entire process highly efficient.
> > >
> > > m = 1e5, A100 40G:
> > > |IPC|1|5|10|20|50|
> > > |-|-|-|-|-|-|
> > > |time(s)|0.3901±0.0057|1.0288±0.0092|1.8352±0.0012|3.506±0.0046| 8.5252±0.0055|
> > >
> > > For **motivation**, we propose an efficient sampling design based on the mapped high-normality property. As discussed in Lines 260-270 left column, we need to generate n latents (corresponding to n IPC) probabilistically following the distribution of the whole class  from Sec. 4.3. Therefore, Gaussian sampling is directly used to approximate the target distribution. Although each sample is randomly sampled following the target distribution, the overall distribution of all samples may still deviate from the target due to the limited number n of all samples. Thus, to mitigate this issue, we propose the group sampling method in Sec. 4.4. .
> > >
> > > Specifically, as smaller sample sizes lead to higher statistical bias, we propose group sampling to generate multiple subsets and select the most representative one with the closest distribution to the target. Thus, although one subset may suffer from large statistical bias, it is possible to find another subset with smaller bias through our group sampling. The results of ablation study for group sampling  in Table 3 and the discussions in Line 357-368 right column show our best performance compared with baselines, demonstrating the effectiveness of the sampling method. The motivation for our group sampling is straightforward and the performance is outstanding.
> > >
> > > For **optimality**, as discussed above, the group sampling method is very efficient and effective. It only needs several seconds to sample 1e5 subsets and select the best one, and results in the best performance compared with SOTA baselines. Reviewer mentions a possible analytical solution to sample n-3 examples and solve the rest examples to match the target distribution accurately, achieving the optimality to some extent. However, it may require additional complex solvers with high computation cost. Furthermore, with the analytically solved examples, although the statistics of the distribution may match perfectly, the overall distribution of all samples may not be a Gaussian distribution. It is possible that the designed/solved examples are far away from the randomly sampled examples to match the target distribution which is in the middle between them.

---

### Decision · Program_Chairs · 2025-05-01

**Decision:**

Accept (poster)

**Comment:**

The submission has been reviewed by 5 reviewers.

After the rebuttal, all reviewers acknowledged the merits of the manuscript and agreed that it meets the bar of ICML.

Essentially, the reviewers found the proposed method makes sense and gives rise to good results, among others.

There is no basis to overturn the consensus. As such, the AC recommends the acceptance. Congrats!

Please, however, do account for the suggestions in the final version.